# PSBench: a large-scale benchmark for estimating the accuracy of protein complex structural models

**Pawan Neupane**[*]
University of Missouri - Columbia
`pngkg@missouri.edu`

**Jian Liu**[*]
University of Missouri - Columbia
`jl4mc@missouri.edu`

**Jianlin Cheng**[†]
University of Missouri - Columbia
`chengji@missouri.edu`

## Abstract

Predicting protein complex structures is essential for protein function analysis, protein design, and drug discovery. While AI methods like AlphaFold can predict accurate structural models for many protein complexes, reliably estimating the quality of these predicted models (estimation of model accuracy, or EMA) for model ranking and selection remains a major challenge. A key barrier to developing effective machine learning-based EMA methods is the lack of large, diverse, and well-annotated datasets for training and evaluation. To address this gap, we introduce PSBench, a benchmark suite comprising five large-scale, labeled datasets, four of which were generated during the 15th and 16th community-wide Critical Assessment of Protein Structure Prediction (CASP15 and CASP16), and one curated for new Protein Data Bank (PDB) entries deposited between July 2024 and August 2025. PSBench includes over 1.4 million structural models covering a wide range of protein sequence lengths, complex stoichiometries, functional classes, and modeling difficulties. Each model is annotated with multiple complementary quality scores at the global, local, and interface levels. PSBench also provides multiple evaluation metrics and baseline EMA methods to facilitate rigorous comparisons. To demonstrate PSBench's utility, we trained and evaluated GATE, a graph transformer-based EMA method, on the CASP15 data. GATE was blindly tested in CASP16 (2024), where it ranked among the top-performing EMA methods. These results highlight PSBench as a valuable resource for advancing EMA research in protein complex modeling. PSBench is publicly available at: `https://github.com/BioinfoMachineLearning/PSBench`.

## 1 Introduction

Proteins are essential biological macromolecules whose diverse functions in living organisms are dictated by their three-dimensional (3D) structures. Although experimental techniques such as X-ray crystallography, cryo-electron microscopy (cryo-EM), and nuclear magnetic resonance (NMR) spectroscopy can determine protein structures with high accuracy, these approaches are time-consuming and resource-intensive and can only be applied to a tiny portion ($< 0.1\%$) of proteins.

To overcome these challenges, machine learning methods[1, 2, 3, 4, 5, 6, 7, 8, 9] for predicting protein structures from sequences have become essential. Among these, AlphaFold[1, 10, 11] has

---

[*]Equal contribution.
[†]Corresponding author.

39th Conference on Neural Information Processing Systems (NeurIPS 2025) Track on Datasets and Benchmarks.

revolutionized the field by achieving experimental accuracy for predicting the tertiary structures of almost all single-chain proteins (monomers) first and then delivering high-accuracy structure prediction for a large portion of multi-chain protein complexes (multimers). As monomer structure prediction is largely considered solved, protein complex structure prediction is currently one major focus in the field. Despite its success, a critical limitation persists: AlphaFold's self-estimated model accuracy (quality) scores (e.g., plDDT, pTM, ipTM, and confidence scores) are not always reliable for identifying high-quality predicted complex structures (structural models)[12]. For instance, using AlphaFold2-Multimer or AlphaFold3 to predict many (e.g., thousands of) structural models for a protein complex target can substantially increase the likelihood of generating some high-quality ones[9], but AlphaFold's own confidence scores or ranking scores often cannot rank them at the top when the ratio of high quality models versus low-quality models is low. As a result, selecting structural models of high quality from a pool of models generated by AlphaFold or other AI methods is a major challenge in protein complex structure prediction and sometimes even harder for users than model generation itself[8].

This challenge highlights the importance of Estimation of Model Accuracy (EMA) (also called model quality assessment), which predicts how closely a predicted structural model resembles the native (true) structure before the true structure is known. Reliable EMA tools are not only critical for model selection in the prediction phase, but also vital to prioritize accurate structural models for downstream applications such as protein function annotations and drug discovery. To stimulate the development of EMA methods for protein complex structural models, since 2020, CASP has dedicated one competition category to assess EMA methods[13, 14]. However, EMA methods remain underdeveloped due to two critical gaps: the lack of large, high-quality, labeled complex model datasets to train and test machine learning EMA methods (like ImageNet for image processing) and the lack of user-friendly standardized benchmarks and automated evaluation tools to assess the performance of EMA methods.

To bridge this gap, we introduce PSBench, a comprehensive benchmark for training and testing EMA methods to predict the accuracy (quality) of predicted protein complex structural models and comparing them with baseline methods via multiple complementary metrics (Fig. 1). PSBench consists of five complex structure datasets, including four datasets with more than one million community-predicted and in-house-predicted structures for protein complex targets of the 2022 CASP15[14, 15, 16] and 2024 CASP16 competitions[17], and a newly curated dataset of AlphaFold3-predicted models for the PDB entries deposited between July 2024 and August 2025. The CASP-related datasets were generated in the truly blind prediction setting (e.g., true structures were unavailable during prediction). These structural models were predicted mainly by AlphaFold2-Multimer[10] and AlphaFold3[11] for 79 diverse, representative protein complex targets with different lengths, difficulties and stoichiometries (count of each unique chain in a protein complex), carefully selected by protein structure experts[15]. The newly curated dataset further expands this diversity by adding 400,400 structural models for 2,002 targets, increasing the total number of complex targets to 2,081. Each model is rigorously labeled with 10 distinct quality scores spanning global, local and interface accuracy measures (Fig. 1a). Importantly, CASP15 models and CASP16 models were generated two years apart and therefore can provide a rigorous split of data for training and testing EMA methods, preventing information leakage and mirroring real-world EMA workflows.

To demonstrate PSBench can be used to train advanced EMA methods and rigorously benchmark them prior to their use, we trained and tested GATE[18], a graph transformer-based EMA method on two CASP15 datasets (CASP15_inhouse_dataset and CASP15_community_dataset) respectively for two purposes: (1) estimating the accuracy of structural models generated by one predictor for model selection and ranking, a typical setting for structure predictors and users; and (2) estimating the accuracy of structural models generated by many predictors in a community, which is a typical setting of CASP EMA competition. We then blindly tested two GATE variants in the blind CASP16 competition held from May to August 2024.

In the official CASP16 EMA competition category, GATE ranked among the best methods out of 38 participating EMA predictors. In the blind ranking and selection of in-house structural models predicted by our own protein structure system (MULTICOM4) built on top of AlphaFold2-Multimer and AlphaFold3 during CASP16, GATE also outperformed five standard EMA methods and helped MULTICOM4 rank among top predictors in protein complex structure prediction in CASP16[9]. The results demonstrate that PSBench is a valuable resource, including labeled datasets, a model annotation pipeline, baseline EMA methods, and evaluation tools/metrics, for the AI community to

develop and benchmark cutting-edge machine learning methods to estimate protein complex model accuracy, addressing a significant bottleneck in the field of protein structure prediction.

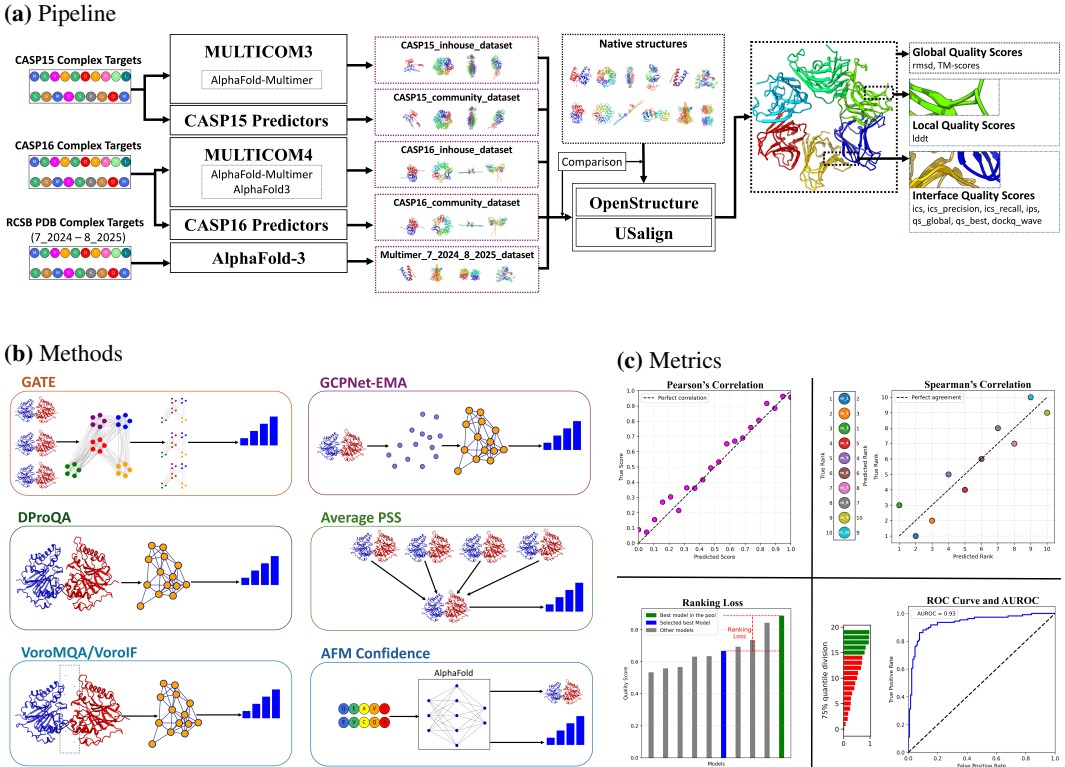

Figure 1: **Overview of PSBench. (a) Pipeline.** The PSBench pipeline for preparing five datasets for estimating protein complex model accuracy (EMA). The predicted structural models are compared with native (true) structures to compute global, local, interface quality scores as labels. **(b) Methods.** Six representative baseline EMA methods for performance comparison. **(c) Metrics.** Four metrics for evaluating EMA methods: Pearson's correlation, Spearman's correlation, ranking loss, and AUROC (Area Under Receiver Operating Characteristics Curve) for evaluating predicted model quality scores against true ones (labels). The evaluation tools are included in PSBench.

## 2  Background and Related Work

The development of EMA methods, either physics-/statistical potential-based methods[19, 20] or data-driven machine learning methods[21, 22, 23], requires the availability of high-quality datasets of predicted protein complex structural models with annotated quality scores. Particularly, the recently emerged, more powerful deep learning-based EMA methods need to be trained and tested on large, diverse structural model datasets to reliably predict the accuracy (quality) of structural models sampled from the vast protein structure space.

Early benchmark datasets, such as the Docking Benchmark (BM)[24], PPI4DOCK[25], and DockGround[26], consist of structural models generated by traditional protein docking tools like SwarmDock[27], ZDock[28], and pyDock[29]. However, as docking tools are not accurate and have been largely replaced by much more accurate AlphaFold, these datasets are not suitable for training machine learning methods to predict the quality of structural models generated by widely used AlphaFold. Moreover, these datasets are relatively small. For instance, PPI4DOCK has 54,000 structural models, and CAPRI Score_set[30] comprises 19,013 structural models for 15 targets. Furthermore, the structural models were mostly predicted for small protein complexes (such as homo- and hetero-dimers), which cannot represent large protein complexes consisting of many chains and having more complicated stoichiometries.

Recent efforts have sought to apply state-of-the-art protein complex structure predictors like Al-phaFold2 and AlphaFold2-Multimer to create benchmark datasets. For instance, Multimer-AF2 Dataset (MAF2)[31] comprises 9,251 structural models generated by AlphaFold2 and AlphaFold2-Multimer, while Heterodimer-AF2 Dataset (HAF2)[31] is a collection of 1,849 structural models for 13 heterodimer proteins generated by the same tools. These datasets have enabled the development of advanced EMA methods such as DProQA[31] and ComplexQA[32]. However, like the early datasets, these datasets contain a small number of structural models generated for small to medium protein complexes (e.g., sequence length less than 1500 residues), and therefore cannot represent the diverse protein complex structure space well. Moreover, both early and recent benchmark datasets have a limited set of quality scores assigned to each structural model without capturing different aspects of model quality. And the structural models in these datasets were generated in a simulated prediction environment where true structures were known, which is different from the truly blind prediction setting where structural models are predicted prior to true structures being available.

Finally, the previous benchmarks do not provide automated evaluation tools and baseline methods to benchmark new EMA methods, which are important for speeding up the development of machine learning EMA methods. And they do not include tools to automatically annotate and incorporate new structural models so that they cannot be expanded.

To address the gaps above, we develop PSBench, a large, comprehensive benchmark for developing, training, and testing EMA methods for protein complex. PSBench makes the following unique contributions to the field.

- Providing more than 1.4 million complex structural models generated by state-of-the-art deep learning methods (mostly AlphaFold2-Multimer and AlphaFold3), much larger than previous datasets.

- Four of the five datasets (about 1 million models) were generated in the real-world blind prediction setting (CASP15 and CASP16 competitions) without any knowledge of true structures. In addition, a newly curated dataset of AlphaFold3-predicted models for protein complexes deposited in the PDB between July 2024 and August 2025 enables continuous benchmarking with recent structural data.

- The structural models were generated for 2,081 diverse protein complex targets including 79 CASP ones carefully selected by protein structure experts, encompassing 185 distinct stoichiometries, multiple functional classes, varying difficulty levels (easy, medium, and hard), and a wide range of sequence lengths (12 to 8,460 residues).

- The structural models are assigned 10 complementary quality scores at the local, global, and interface levels, measuring their accuracy from different perspectives, important for training and evaluating EMA methods.

- Providing automated evaluation tools for comparing new EMA methods with 6 standard baseline EMA methods, as well as a model annotation (labeling) pipeline for continuous expansion of the datasets.

- The utility of PSBench for developing state-of-the-art EMA methods was blindly and rigorously proved in CASP16.

## 3   PSBench Design

PSBench encompasses over 1.4 million predicted structural models, distributed across five datasets: CASP15_inhouse_dataset, CASP15_community_dataset, CASP16_inhouse_dataset, CASP16_community_datase, and Multimer_7_2024_8_2025_dataset. The first four datasets were generated for 79 CASP complex targets during the 2022 CASP15 competition and the 2024 CASP16 competition, while the fifth dataset consists of AlphaFold3-generated structural models for 2,002 non-redundant multimeric protein entries deposited in the RCSB PDB between July 2024 and August 2025 (Fig. 1a).These targets represent 185 distinct stoichiometries (Fig. S1a in Appendix A.1) and more than 145 protein classes (Fig. S1b), providing a broad coverage of protein complexes.

The structural models were compared with the corresponding native (true) structures of the targets by an automated annotation pipeline in PSBench to assign quality scores to them as labels. The annotation pipeline pre-processed structural models and true structures so that they could be aligned and compared by two tools, OpenStructure[33, 34, 35] and USalign[36], generating 10 complementary

quality scores measuring model accuracy from three different aspects: global quality, interface quality, and local quality (see the detailed description of these quality scores in Appendix A.2). The global quality scores (variants of `tm-score` and `rmsd`) quantify the similarity between the global fold of a model and that of the true structure. The interface quality scores (`ics`, `ics_precision`, `ics_recall`, `ips`, `qs_global`, `qs_best`, and `dockq_wave`) measure the quality of interface regions where two chains in a protein complex interact. The local quality score (`lddt`) measures the accuracy of the location of each residue with respect to its contacted residues.

The quality scores computed by the annotation pipeline for the models in the CASP15_community_dataset and CASP16_community_dataset were cross-validated with the scores compiled from the CASP15 and CASP16 websites to make sure that it worked correctly (see Appendix A.3 for preprocessing, implementation details and edge cases). Users can use one or more quality scores to train and test their EMA methods. Generally, it is recommended at least one global quality score and one interface quality score be used to benchmark EMA methods. We also analyzed inter-score relationships and redundancy (Appendix A.4) and examined how the interface contact density relates to interface quality scores (Appendix A.5). A detailed guidance linking each quality score to relevant structural biology and bioinformatics applications is presented in Appendix A.6. The main characteristics of the five datasets are discussed below.

## 3.1 CASP15_inhouse_dataset

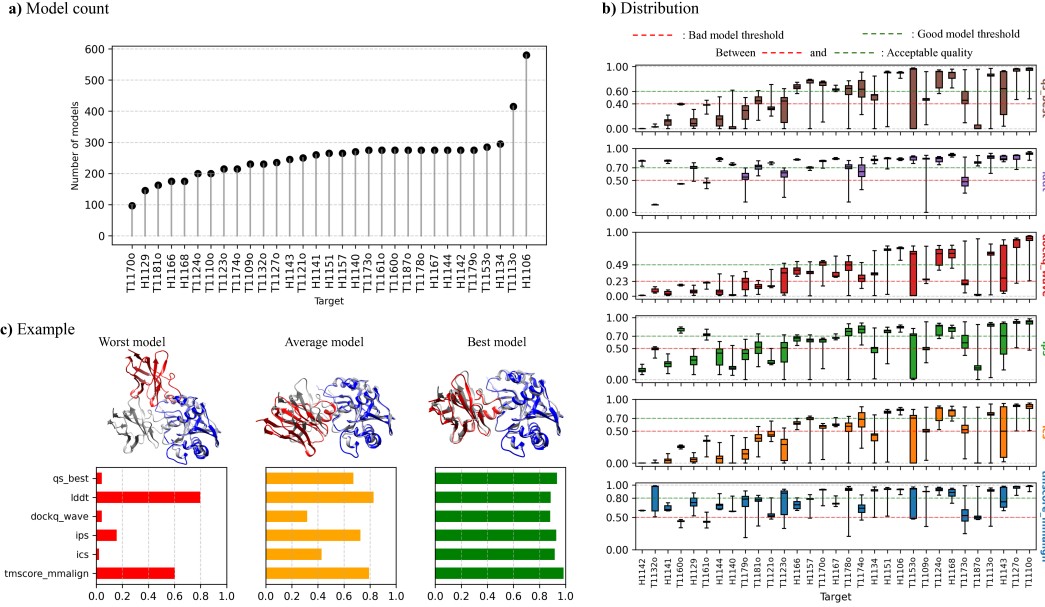

**Figure 2: CASP15_inhouse_dataset. (a) Model count.** Number of models per target in the dataset. **(b) Score Distribution.** Box plots of each of six representative quality scores of the models for each target. **(c) Example.** Three representative models (worst, average, best) in terms of sum of the six representative quality scores for a target H1143. Each model with two chains colored in blue and red is superimposed with the true structure in gray.

The structural models in CASP15_inhouse_dataset were generated by our MULTICOM3[8] system during the 2022 CASP15 competition. MULTICOM3 generated diverse multiple sequence alignment (MSA), structural templates and hyperparameters as input for AlphaFold2-Multimer[10] to predict the structures for 31 CASP15 complex targets and ranked as one of top 10 complex predictors in CASP15. It generated 97 to 580 structural models per target (see model count per target in Fig. 2a), resulting in 7,885 models in total included in CASP15_inhouse_dataset.

The 31 CASP15 targets cover a wide range of distinct stoichiometries, sequence lengths, protein classes (see Table S1 for details). The distribution of six representative quality scores of the structural models for the targets are visualized as box plots in Fig. 2b. It can be seen that the targets are of different difficulty levels. Some easy targets have most models above the threshold of good quality,

some average targets have one portion of models above the good model threshold and another portion below the bad model threshold, and some difficult targets have most models below the bad model threshold. Therefore, this dataset is ideal for benchmarking EMA methods' capability to work for different kinds of targets and train them to possess the capability. The detailed numbers of good, acceptable, and bad models are reported in Table S2. Fig. 2c illustrates three representative models (worst, average, and best) of a target H1143 and their quality scores.

Because the structural models in CASP15_inhouse_dataset were generated by our own predictor, in addition to a structural file in the Protein Data Bank (PDB) format, every model has four extra features, i.e., four estimated quality scores assigned by AlphaFold2-Multimer during the model generation process, including AlphaFold2-Multimer confidence score (`afm_confidence_score`), interface predicted Template Modeling score (`iptm`), number of inter-chain predicted aligned errors (<5 Å) (`num_inter_pae`), and predicted multimer DockQ score (`mpDockQ`) (see Appendix A.7 for details). This is different from CASP15_community_dataset generated by many CASP15 predictors that did not submit these features. It is worth noting that these estimated model quality scores are not true quality scores, but can be used as input features to improve the prediction of true quality scores[18]. Therefore, CASP15_inhouse_dataset is an excellent resource for training EMA methods to rank and select structural models generated by AlphaFold-based structure predictors.

### 3.2 CASP15_community_dataset

The CASP15_community_dataset contains structural models submitted by 87 predictors during the CASP15 competition, where each predictor submitted at most 5 models for each target. Most predictors used different variants of AlphaFold2 and AlphaFold2-Multimer to generate models, while some predictors also used other structure prediction methods such as template-based modeling and protein language model-based modeling (e.g., ESMFold[37]), resulting in a diverse set of structural models. It contains 215 to 319 structural models per target and 10,942 models in total for 40 complex targets (see model count per target in Fig. S5a and Table S3).

Unlike CASP15_inhouse_dataset generated in an in-house controlled generation process, CASP15_community_dataset has more variability in modeling approaches and model quality. As shown in Fig. S5b, the quality scores of the models for many targets spread in a wider range, making the dataset particularly valuable for training and benchmark EMA methods for estimating the accuracy of the models generated by a diverse set of predictors with different performance. Fig. S5c illustrates three representative models for a target. Table S4 reports the number of bad, acceptable and good models for each target. Some targets such as T1160o have only one or a few good models, making them challenging targets for EMA methods to pick good ones.

### 3.3 CASP16_inhouse_dataset

The structural models in CASP16_inhouse_dataset were generated by our MULTICOM4 system[9] built on top of both AlphaFold2-Multimer and AlphaFold3[11] during 2024 CASP16 competition. MULTICOM4 ranked no. 1 in the Phase 0 competition of CASP16 in which the stoichiometry of each target was not provided and needed to be predicted and among top 5 in the Phase 1 competition in which the stoichiometry was provided. MULTICOM4 generated 712 to 78,410 structural models for each target (see model count per target in Fig. S6a), resulting in 1,009,050 models for 36 complex targets. The stoichiometry, total number of models, number of AlphaFold3 models, sequence length, and protein class for each target are reported in Table S5. The CASP16 targets represent a broad range of stoichiometries, protein classes, and sequence lengths.

Fig. S6b illustrates the distribution of the six representative quality scores of the models across targets. In contrast to CASP15_inhouse_dataset, CASP16_inhouse_dataset exhibits greater variability in model quality, due to a much larger number of models per target and the use of both AlphaFold2-Multimer and AlphaFold3. Table S6 reports the number of good, acceptable, and bad models for each target. Some targets have many good models, while other have very few, representing different levels of difficulty. As an example, the worst, average, and best models of target T1235o are shown in Fig. S6c.

Same as CASP15_inhouse_dataset, each model in CASP16_inhouse_dataset also includes four AlphaFold2-Multimer-like self-estimated quality features. Additionally, AlphaFold3-based models have one more feature, `af3_ranking_score`. These scores can be used as input features for EMA

methods to predict the quality of the models. Due to these additional features and a very large number of structural models for a diverse set of protein complex targets, CASP16_inhouse_dataset is a large, valuable resource to train and benchmark EMA to estimate the accuracy of structural models predicted by both AlphaFold2-Multimer and AlphaFold3. Particularly, to our knowledge, it contains the largest number of labeled protein complex structural models generated by AlphaFold3 to date.

### 3.4 CASP16_community_dataset

CASP16_community_dataset comprises 12,904 models for 39 complex targets (166 to 377 models per target) generated by 82 predictors during the CASP16 competition. Most of the predictors used AlphaFold2-Multimer and/or AlphaFold3 to generate structural models, even though some predictors used additional prediction techniques. The per-target model counts, distribution of quality scores and three model examples are illustrated in Fig. S9. The stoichiometry, protein class, sequence length, and number of good/acceptable/bad models of each target are reported in Tables S7 and S8.

Like CASP15_community_dataset, the models in CASP16_community_dataset originated from a large number of diverse predictors and therefore have a wide range of quality scores (Fig. S9b). The dataset is ideal for training and benchmarking EMA methods for predicting the quality of structural models generated by various modern deep learning-based protein structure prediction methods including, but not limited to AlphaFold2-Multimer and AlphaFold3.

### 3.5 Multimer_7_2024_8_2025_dataset

Multimer_7_2024_8_2025_dataset expands the CASP datasets by including 400,400 AlphaFold3-predicted models (200 per target) for 2,002 non-redundant multimeric protein entries deposited in the RCSB PDB between July 2024 and August 2025. Initially, 3,379 targets were collected from the PDB and subjected to redundancy reduction by grouping entries that contain the same number of protein chains with identical sequences but differ in ligands or nucleic acids, followed by selection of representative targets (i.e. the target in each group that appeared first in the PDB). This process yielded 2,216 unique entries for AlphaFold3 prediction, of which 214 could not be processed due to GPU memory constraints arising from large residue count. It is worth noting that the PDB is continuously updated, and the exact number of available entries within a deposition window may vary over time.

This dataset represents the most diverse collection in PSBench, encompassing 179 distinct stoichiometries and 143 distinct protein classes. The lengths of the targets range from 12 to 4,591 residues (see Fig. S10). The distribution of average structural quality scores across all targets in the `Multimer_7_2024_8_2025_dataset` is shown in Fig. S11.

Similar to the in-house datasets, each model in `Multimer_7_2024_8_2025_dataset` contains AlphaFold-derived self-estimated features, including `afm_confidence_score` ( Fig. S12), `af3_ranking_score` (Fig. S13), `iptm`, `num_inter_pae`, and `mpDockQ`. These features can be used as input for EMA methods to predict structural accuracy.

## 4 Evaluation Framework

### 4.1 Evaluating the Utility of PSBench for Training and Testing EMA Methods

To assess if PSBench could support the development of machine learning EMA methods, we trained and validated a graph transformer EMA method (GATE[18]) on CASP15_inhouse_dataset and CASP15_community_dataset separately to obtain two EMA predictors, i.e., (1) one (referred to as GATE-AFM) for predicting the quality scores of structural models generated by AlphaFold and (2) another (referred to as GATE) for predicting the quality scores of structural models generated by many predictors participating in CASP. The two predictors were blindly tested during the CASP16 competition from May to August 2024 as follows.

First, GATE-AFM was used to predict the quality of the in-house structural models generated by MULTICOM4 and select top ones to submit to CASP16 for complex structure prediction competition. To benchmark how GATE performed, five standard EMA methods with source codes available were blindly run in parallel during CASP16, which included DProQA[31], VoroIF-GNN scores[38], GCPNet-EMA[39], PSS[40] and AlphaFold2-Multimer confidence scores (AFM Confidence) (Fig.

1b) (see Appendix B.1 for details). The model quality scores predicted by GATE-AFM and the five methods were compared with the true quality scores available only after CASP16 concluded in December 2024 (see the results in Section 5.1). The six methods are included in PSBench for comparison with future EMA methods.

Second, GATE directly participated in the EMA competition category of CASP16 to evaluate the complex structural models generated by CASP16 complex structure predictors. GATE was assessed along with 37 EMA predictors participated in the CASP16 EMA competition by CASP16 organizers and assessors (see the results in Section 5.2). This assessment is highly rigorous and objective because GATE was evaluated with the best EMA predictors in the field by external experts.

## 4.2 Protocols of Training and Validating GATE-AFM and GATE on CASP15 Datasets

GATE-AFM and GATE use the same graph transformer architecture to predict the global quality scores of structural models. It takes as input a graph of a set of structural models of a target, in which a node denotes a model and an edge connects two similar models, to predict the quality score (e.g., TM-score) of each model. The common features for each node shared by both GATE-AFM and GATE are the estimated model quality scores assigned by several EMA methods. The only difference between GATE-AFM and GATE is that the former uses four additional AlphaFold2-Multimer features (confidence scores, ipTM, number of inter-chain predicted aligned errors and mpDockQ) that are only available in CASP15_inhouse_dataset but not in CASP15_community_dataset. GATE-AFM and GATE use the same set of edge features, including structural similarity scores between two connected nodes.

GATE was trained and validated on the CASP15_community_dataset via 10-fold cross-validation, which comprises 10,935 models of 40 complex targets, plus 187 models from another target (i.e., T1115o) whose native structure is not publicly available. For this target, we obtained its quality scores from the CASP15's website as labels. For each target, a pairwise similarity graph was constructed for all the models of each target first. 2000 subgraphs containing up to 50 nodes were then sampled from the full graph of each target to train and validate GATE. GATE-AFM was trained and validated on the CASP15_inhouse_dataset in the similar way. To reduce computational costs, only a subset of CASP15_inhouse_dataset was used to train GATE-AFM (see Appendix A.13). The details of training and validation are provided in the Appendix B.2, and a discussion of computational requirements is included in Appendix B.3.

### 4.3 Evaluation Metrics

PSBench provides four complementary metrics (Fig. 1c) to evaluate the performance of EMA methods. Pearson's correlation coefficient measures the linear correlation between predicted quality scores from an EMA method and ground-truth quality scores. Spearman's correlation coefficient evaluates rank-order consistency between predicted scores and ground-truth scores. Ranking loss directly assesses model selection capability by computing the difference between the ground-truth quality score of the truly best model (highest ground-truth score) and the ground truth score of the no.1 model selected by an EMA method, where lower loss values correspond to better selection and 0 means a prefect selection. Area Under the Receiver Operating Characteristic Curve (AUROC) quantifies binary classification performance by labeling models as high-quality (above the 75[th] percentile of ground-truth scores) or low-quality otherwise. An AUROC of 1.0 indicates perfect classification, while 0.5 corresponds to random guessing. Each metrics is usually calculated for the models of each protein target first and then is averaged over all the targets in a dataset as the performance score for an EMA method. The scripts that can automatically calculate these scoring metrics for EMA methods are included in PSBench.

## 5 Results

### 5.1 Blind Prediction Results of Estimating the Accuracy of CASP16 In-house Models

During CASP16 competition, GATE-AFM pretrained on a subset of CASP15_inhouse_dataset, was blindly applied to our in-house models generated by MULTICOM4. Due to the three-day prediction time constraint, we used it to predict the quality of only hundreds of top-ranked models for each target (i.e., the collection of top 5 models generated by each of dozens of predictors based on AlphaFold2-

Multimer and AlphaFold3 in MULTICOM4). Four targets (T1249v1o and T1249v2o, T1294v1o and T1294v2o) that have the same sequence but different conformations are excluded. The performance of GATE and the other EMA methods on the top models for the remaining 32 targets (referred to as CASP16_inhouse_TOP5_dataset, a subset of CASP16_inhouse_dataset) was compared in Table 1.

**Table 1:** Performance of EMA methods in estimating the accuracy of the CASP16 in-house models. Metrics include Pearson's correlation (Corr$^P$), Spearman's correlation (Corr$^S$), ranking loss, and AUROC, reported separately for TM-score and DockQ_wave. Bold font and underline denote the best and second best results respectively. Significant difference ($p < 0.05$) between GATE-AFM and other methods based on the one-sided Wilcoxon signed-rank test is marked with *.

| Method | TM-score | | | | DockQ_wave | | | |
|---|---|---|---|---|---|---|---|---|
| | Corr$^P$ ↑ | Corr$^S$ ↑ | Loss ↓ | AUROC ↑ | Corr$^P$ ↑ | Corr$^S$ ↑ | Loss ↓ | AUROC ↑ |
| GATE-AFM | 0.372 | **0.283** | **0.102** | **0.658** | **0.431** | **0.322** | **0.138** | **0.662** |
| AFM Confidence | 0.259* | 0.143* | 0.106 | 0.597* | 0.252* | 0.114* | 0.151 | 0.593* |
| PSS | **0.394** | 0.261 | 0.114 | 0.647 | 0.369 | 0.284 | 0.154 | 0.645 |
| GCPNet-EMA | 0.360 | 0.249 | 0.135 | 0.643 | 0.355 | 0.264 | 0.169 | 0.648 |
| VoroMQA-dark | 0.039* | 0.144 | 0.129 | 0.609 | -0.013* | 0.146* | 0.163 | 0.622 |
| VoroIF-GNN-pCAD-score | 0.073* | 0.105* | 0.167* | 0.589* | 0.074* | 0.137* | 0.204 | 0.615 |
| VoroIF-GNN-score | 0.065* | 0.116* | 0.193* | 0.599* | 0.114* | 0.170* | 0.207* | 0.622 |
| DProQA | -0.051* | 0.011* | 0.194* | 0.569* | 0.032* | 0.071* | 0.223* | 0.587* |

GATE-AFM outperformed the other methods according to almost all evaluation metrics. In terms of a global quality score - TM-score, GATE-AFM achieved the highest Spearman's correlation (0.283), the lowest ranking loss (0.102), the best AUROC (0.658), and second highest Pearson's correlation (0.372), indicating superior ranking consistency and classification reliability. In terms of an interface quality score - DockQ_wave, GATE-AFM again outperformed all other methods, attaining the highest Pearson's correlation (0.431), the highest Spearman's correlation (0.322), the lowest ranking loss (0.138), and the highest AUROC (0.662). It performed better than AFM Confidence (the default self-estimated quality score of AlphaFold2-Multimer) in terms of all the metrics, demonstrating its significant value of estimating the quality of AlphaFold-generated structural models. In many cases, the improvement of GATE-AFM over the other methods is significant (see the cases marked with * in Table 1). Additional AUROC analyses with fixed thresholds (e.g., TM-score $\geq 0.5$, DockQ_Wave $\geq 0.49$) are provided in Appendix B.4. To complement these aggregate results, we also examined some failed cases with large prediction errors, which revealed two distinct failure modes (Appendix B.5). GATE-AFM's strong capability of selecting good models was one important reason that our MULTICOM4 predictors ranked among top predictors in the CASP16 complex structure prediction category[9]. These results show that PSBench can be used to develop state-of-the-art EMA methods.

## 5.2 Blind Prediction Results in CASP16 EMA Competition

GATE, pretrained on CASP15_community_dataset, participated in 2024 CASP16 EMA competition under the predictor name: MULTICOM_GATE. We downloaded the EMA prediction results of MULTICOM_GATE and other 37 CASP16 EMA predictors from the CASP16 website. Two very large targets (H1217 and H1227) were excluded because GATE did not generate predictions for their models due to time constraints during CASP16, resulting in using 37 out of 39 targets in CASP16_community_dataset for evaluation.

The blind prediction results of top 20 out of 38 CASP16 EMA predictors were shown in Fig. 3. In terms of TM-score, MULTICOM_GATE was ranked first according to Pearson's correlation (0.673), third according to Spearmans' correlation (0.456), ranking loss (0.135) and AUROC (0.652) respectively. Moreover, we assessed the performance of the 38 EMA predictors using a Z-score–based ranking, where MULTICOM_GATE ranked third (see detailed results in Appendix B.6). The outstanding performance of MULTICOM_GATE in CASP16 EMA competition[41] highlights PSBench's unique value: its high-quality training and test datasets and rigorous evaluation protocols provide a strong framework to develop and validate state-of-the-art machine learning methods.

## 6 Conclusion, Limitations and Future Work

PSBench fills critical gaps in protein complex model accuracy estimation by providing a large-scale benchmark with both data and tools for training and testing EMA methods. It contains more than one million labeled structural models for 79 CASP15/16 targets that cover diverse stoichiometries, function classes, sequence lengths, and difficulty levels. PSBench's value of supporting the development of generalizable EMA methods has been demonstrated by the outstanding performance of GATE trained with it in the blind community-wide CASP16 competition in 2024.

Since CASP16, PSBench has been substantially expanded by including the structural models of 2,002 new non-redundant targets deposited in the PDB between July 2024 and August 2025. We will continue to ex-

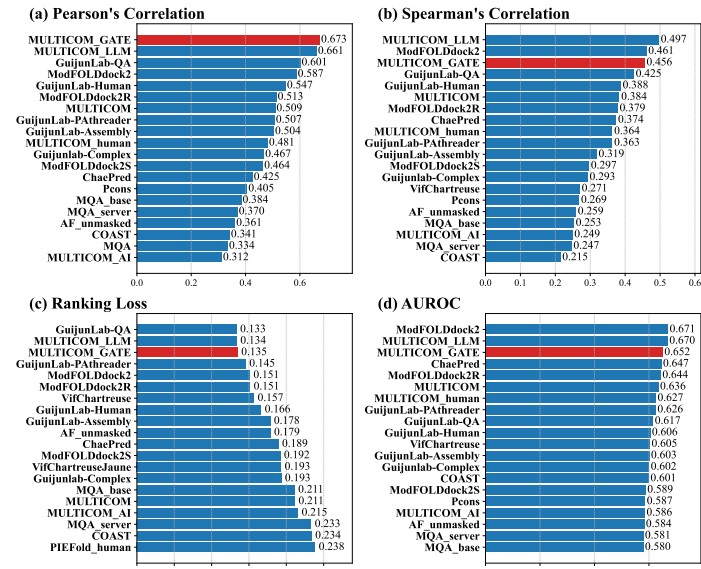

**Figure 3: CASP16 EMA results.** The performance of top 20 out of 38 CASP16 EMA predictors in predicting TM-scores of structural models of 37 complex targets. **(a)** Pearson's correlation. **(b)** Spearman's correlation. **(c)** Ranking loss. **(d)** AUROC. MUL-TICOM_GATE is highlighted in red. It ranked first, third, third, and third in terms of the four metrics respectively.

tend PSBench by incorporating structural models for more targets (e.g., the targets of the upcoming 2026 CASP17 competition and newly released protein complex structures in the PDB). Most models in PSBench were generated by AlphaFold, reflecting its current dominance but also introducing potential method-specific bias (see Appendix A.14 for detailed analysis). As new structure prediction methods emerge, we will incorporate their models into PSBench to further expand the diversity of prediction approaches represented in PSBench. We have also provided the model annotation pipeline in PSBench for third-party users to automatically label structural models generated in their research. New quality metrics that emerge in the field will be incorporated into PSBench to ensure the benchmark remains comprehensive and up to date. In addition, we benchmarked the runtime and memory usage of the baseline EMA methods (see Appendix B.7), which highlights the trade-off between the higher accuracy of multi-model EMA approaches such as GATE and the greater efficiency of lightweight single-model EMA methods. Finally, we provide a web server to accept structural models contributed by third-parties and will acknowledge their contribution in the future release of PSBench. Our goal is to make PSBench a community-driven resource like ImageNet or MNIST to support AI researchers to solve the critical protein model accuracy estimation problem.

# 7 Data and Software Availability

**Data Availability**
The PSBench datasets are publicly available at Harvard Dataverse: `https://dataverse.harvard.edu/dataset.xhtml?persistentId=doi:10.7910/DVN/75SZ1U`. DOI: `https://doi.org/10.7910/DVN/75SZ1U`.

**Software Availability**
The programs to evaluate EMA methods on the benchmark datasets, to generate labels for new datasets, and a web server for third-party model upload are available at GitHub: `https://github.com/BioinfoMachineLearning/PSBench`. The requirements to run PSBench are described in Appendix C.

# 8 Acknowledgment

We would like to thank the CASP organizers and community for sharing CASP15 and CASP16 data and NSF (award #: DBI2308699)) and NIH (award #: R01GM146340) for supporting this research.

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

# A  Dataset Design

## A.1  Diversity of Protein Complex Targets in PSBench

**(a)** Stoichiometry                                     **(b)** Protein Class

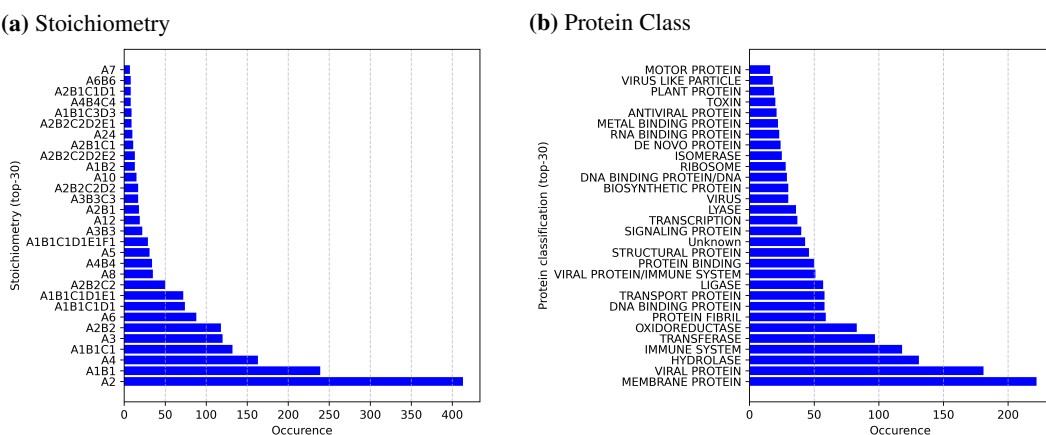

**Figure S1: Diversity of 2081 protein complex targets. (a)** Number of targets for each of top-30 stoichiometries (out of 185) represented in PSBench. A stoichiometry is denoted by letters interleaved with numbers. Each letter represents a unique chain. The number following a letter is the number of the copies (count) of the chain. For instance, A1B2 means a complex has two unique chains A and B, while A has one copy and B has two copies. **(b)** Number of targets for each of top-30 broad protein function classes (out of 145) and an "Unknown" class in PSBench. "Unknown" means there is no class information and therefore may include many different classes.

## A.2  Definitions of Quality Scores

PSBench provides a comprehensive list of quality scores for each protein complex structural model as labels, which are described below.

**Global quality scores**

- **tmscore** : The Template Modeling score (TM-score) measures structural similarity between a predicted structure and a reference structure, with higher values (above 0.8) indicating strong agreement. In our evaluation, four variants of the TM-score are used: **(1)** `tmscore_mmalign`, computed using OpenStructure with the USalign plugin and parameters `-mm 1 -ter 0`, following the CASP16 evaluation protocol; **(2)** `tmscore_usalign`, calculated with the USalign program using parameters `-ter 1 -TMscore 6`, aligning with the CASP15 evaluation protocol; **(3)** `tmscore_usalign_aligned`, which further incorporates residue-residue correspondence via an in-house alignment and filtration script before applying USalign with the same parameters; and **(4)** `tmscore_usalign_aligned_v0`, an earlier version based on a prior alignment script, used for generating GATE EMA training labels and available only for the CASP15_inhouse_dataset. Despite slight procedural differences, all variants apply a consistent threshold interpretation for assessing structural similarity.

- **rmsd** : The Root Mean Square Deviation (`rmsd`) measures the average distance between corresponding atoms in the model and target structures. It quantifies how much the predicted model deviates from the native structure, with lower values indicating a more accurate structure.

**Local quality score**

- **lddt** : The measures the agreement between inter-atomic distances in the model and the target structure. It evaluates how accurately the overall atomic arrangement is reproduced, focusing on all residues from all chains.

**Interface quality scores**

- **ics** : The Interface Contact Score(`ics`) measures how accurately predicted residue contacts between protein chains match the true contacts in the target structure. It is a weighted average of F1-scores for each chain-chain interface, where interfaces with more true contacts contribute more to the overall score.
- **ics_precision** : The Interface Contact Precision(`ics_precision`) measures how many predicted residue contacts are correct, focusing on prediction accuracy rather than coverage. It is a weighted average of precision for each chain-chain interface, with larger interfaces (more true contacts) contributing more to the final score.
- **ics_recall** : The Interface Contact Recall(`ics_recall`)measures how many true residue contacts are correctly predicted, focusing on coverage. It is a weighted average of recall for each chain-chain interface, with larger interfaces (more true contacts) contributing more to the final score.
- **ips** : The Interface Patch Similarity(`ips`) measures the similarity between predicted and true interface residue contacts using the Jaccard coefficient. It is a weighted average of the Jaccard index for each chain-chain interface, with larger interfaces (more true contacts) contributing more to the final score.
- **qs_global** : The QS (global) score(`qs_global`) measures the fraction of correctly predicted interface contacts relative to the total number of true or predicted contacts, whichever is larger. It reflects the overall accuracy of contact prediction across the entire protein complex.
- **qs_best** : The QS (best) score(`qs_best`) measures the highest fraction of correctly predicted interface contacts for any single chain-chain interface in the complex. It highlights the best-performing interface prediction within the entire structure.
- **dockq_wave** : The DockQ_wave(`dockq_wave`) measures the weighted average of DockQ scores across all chain-chain interfaces in the complex. It provides an overall measure of interface prediction quality, combining precision, recall, and Fnat into a single score.

### A.3   Preprocessing, Implementation Details, and Edge Cases of the Annotation Pipeline

To enable fair and reproducible benchmarking of EMA methods, PSBench employs an automated annotation pipeline to compute quality scores for structural models using their corresponding native structures as references. This pipeline ensures that models from diverse sources are consistently aligned, compared, and labeled with multiple complementary accuracy measures.

**Preprocessing.** All native PDB structures were reindexed to match the full-length protein sequences. Non-protein components (e.g., ligands and metal ions) were excluded. If there are insertion codes (e.g., 85A, 86B) in the residue indices of the native structure, they are replaced with indices that correspond to the residue positions in the FASTA sequence. All subsequent residues are renumbered to ensure continuous, monotonic indexing without gaps or duplicates.

**Alignment and quality score computation.** After preprocessing, the pipeline aligns predicted and native structures based on sequence identity and computes scores such as RMSD and lDDT without requiring additional preprocessing. During alignment/evaluation, residues that are missing in the native PDB structure are automatically excluded by OpenStructure, so scoring is restricted to experimentally resolved regions. For TM-score, two routes were used: (i) `tmscore_mmalign`, via OpenStructure's embedded MM-align interface with CASP16-style parameters; and (ii) `tmscore_usalign`, via the standalone US-align program with CASP15-style parameters. For `tmscore_usalign_aligned`, we additionally applied a residue reindexing/filtration step to enforce residue–residue correspondence before running US-align; no such preprocessing was applied to `tmscore_usalign`. TM-score was interpreted following standard practice.

**Edge cases.** Despite the robustness of the pipeline, a small fraction of structural models from the CASP community datasets failed annotation due to format inconsistencies or violations of

OpenStructure's assumptions. For example, non-monotonic residue numbering (e.g., H1272TS191_1 from the `CASP16_community_dataset`) caused alignment failures, while malformed PDB files with duplicate atom labels (e.g., H1114TS229_1 from the `CASP15_community_dataset`) triggered parsing errors. When the number of valid residues in a chain is less than six, OpenStructure refuses to include such chains in chain mapping and evaluation.

**Fallbacks and exclusions.** When OpenStructure failed but US-align succeeded, we retained the model and reported the TM-score from US-align. For example, in the `Multimer_7_2024_8_2025_dataset`, OpenStructure fails for three targets (9DYY, 9KAP, and 9O7J) because all chains contain fewer than six valid residues. For these cases, only TM-scores computed using US-align are included. Models that failed under both frameworks were excluded. For our datasets, such failures were extremely rare, affecting fewer than 0.002% out of more than 1.4 million models.

Finally, it is worth noting that the native structures used as ground truth in PSBench, although experimentally determined, are not error-free. Some structures solved by X-ray crystallography, cryo-EM, or NMR may have some problems such as highly flexible regions, crystal artifacts, or low-resolution density. To mitigate these effects, PSBench limits the evaluation to resolved regions and incorporates local metrics (e.g., lDDT) that are less sensitive to global alignment noise. Users should therefore interpret benchmark results as reflecting both prediction accuracy and the inherent uncertainty of the experimental references.

## A.4 Metric Redundancy Analysis

PSBench provides ten complementary quality metrics that capture different aspects of model accuracy, including global fold, residue-level details, and interfacial quality. While this diversity allows for more comprehensive evaluation, some metrics may be redundant due to overlapping definitions. To better understand their relationships, we analyzed metric redundancy across all five datasets.

The pairwise Pearson's correlation analysis (Figure S2a) indicates moderate to strong dependencies among several metrics. Among the interface contact–based measures, `ics` and `ics_precision` were strongly correlated ($r = 0.88$), and `ics` also showed substantial association with `ics_recall` ($r = 0.74$), indicating that these metrics capture overlapping aspects of interface accuracy. Similarly, `qs_best` and `qs_global` were closely related ($r = 0.76$), consistent with their shared focus on interface quality. `dockq_wave`, which integrates multiple interface-level components such as interface contact accuracy and interface RMSD into a single continuous score, showed moderate correlations with both interface-based metrics (`ics`: $r = 0.64$; `ics_recall`: $r = 0.72$) and global fold metrics (`tmscore_mmalign`: $r = 0.55$). This indicates that `dockq_wave` captures aspects of both local interface geometry and overall model correctness. In contrast, metrics that assess overall or local residue-level accuracy showed weaker correlations with interface-based measures. For example, `lddt` exhibited only moderate correlations with global and interface metrics ($r = 0.56$ with `dockq_wave`; $r = 0.47$ with `tmscore_mmalign`), reflecting its distinct sensitivity to local residue details. `tmscore_mmalign` and `rmsd` also showed moderate inter-metric correlations ($r \approx 0.3$–$0.7$), suggesting complementary perspectives on overall structural similarity.

Principal Component Analysis (PCA) provided further insights into metric relationships as shown in Figure S2b. PC1 exhibited similar negative loadings across nearly all metrics (approximately –0.3), except for `rmsd`, which showed an opposite loading (0.20). This pattern indicates that PC1 captures a general dimension of overall model quality, where improvements in most metrics correspond to lower RMSD values. PC2 was primarily driven by `rmsd` (loading = 0.75), while PC3 was dominated by `lddt` (loading = 0.91). PC4 showed strong contributions from `tmscore_mmalign` (loading = 0.67) and `rmsd` (loading = 0.43). PC5 was characterized by `ips` (loading = 0.64) and `dockq_wave` (loading = -0.57). The variance information (Fig. S2c) shows that PC1 captures the vast majority of variance, with subsequent components contributing progressively less.

To account for redundancy when combining metrics during training, we propose a correlation-based weighting scheme. Each metric is weighted inversely to its average Pearson's correlation with the other metrics, i.e.,

$$w_i = \frac{1}{\frac{1}{N-1} \sum_{j \neq i} r_{ij}},$$

where $r_{ij}$ is the correlation between metrics $i$ and $j$, and $N$ is the total number of metrics. This down-weights highly redundant metrics (e.g., ics, ics_precision, ics_recall) while giving more influence to complementary metrics such as lddt and ips.

The resulting weights can be applied in a weighted loss function to train EMA methods:

$$L(m) = \sum_{i=1}^{N} w_i \, \ell\big(\hat{y}_i(m), y_i(m)\big),$$

where $\ell(\cdot)$ is the loss (e.g., mean squared error) between the predicted score $\hat{y}_i(m)$ and the reference score $y_i(m)$ for metric $i$.

In practice, this means that highly correlated interface metrics contribute less to the final loss, while distinct signals such as those from lddt or rmsd are emphasized. This weighting ensures that training focuses on complementary information rather than duplicating signal from redundant metrics.

**(a)** Quality scores correlation

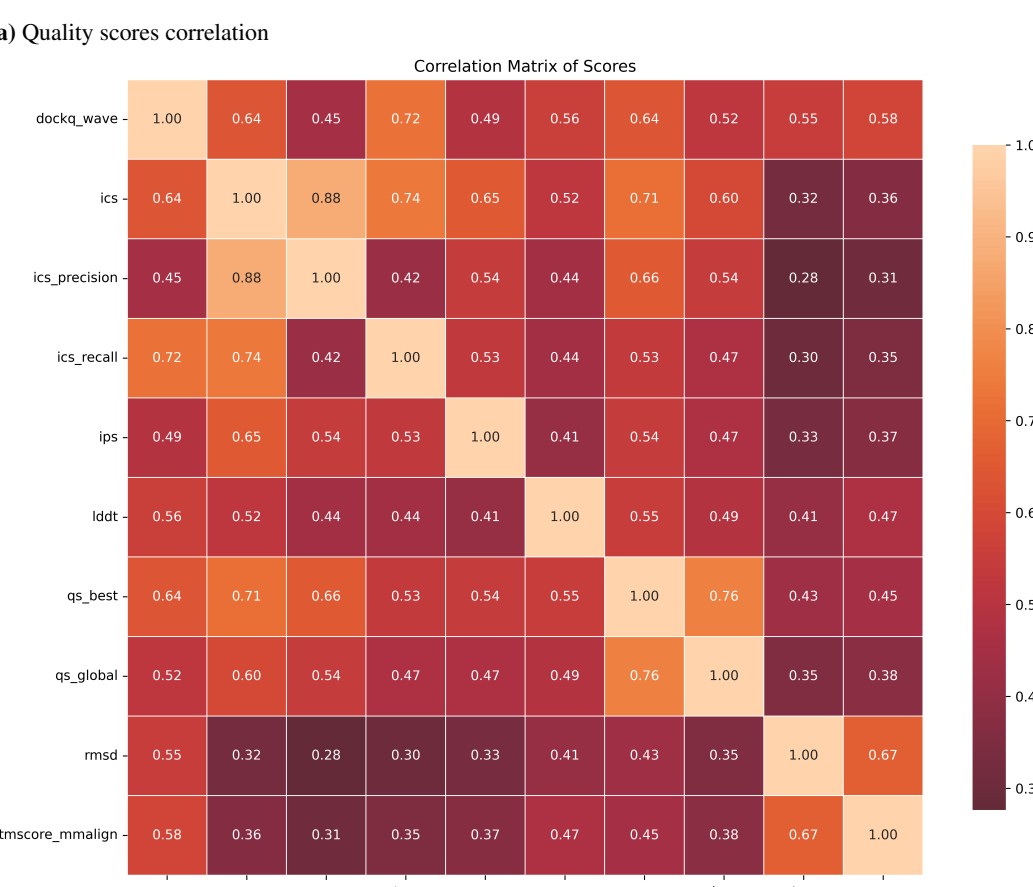

**(b)** Quality score loadings

**(c)** Variance explained

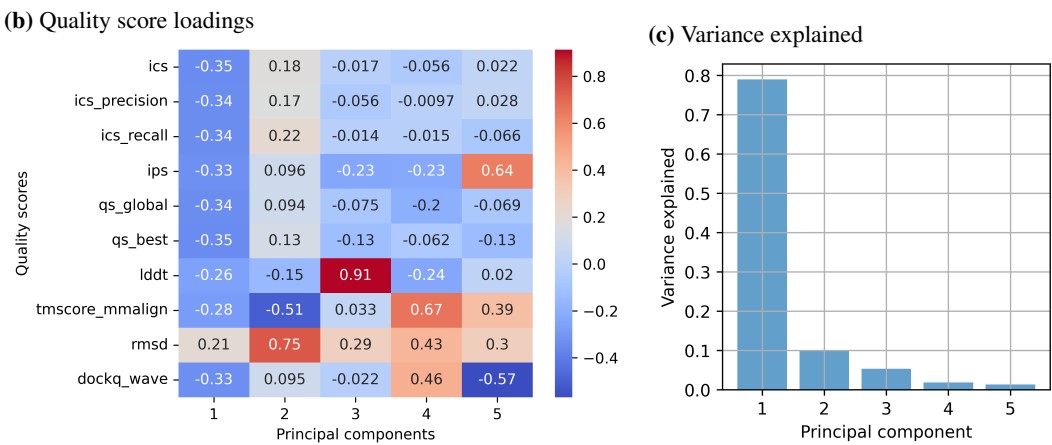

**Figure S2: Relationships among quality scores. (a) Quality scores correlation.** Pairwise correlation between different quality scores. **(b) Quality score loadings.** Loadings of each quality score on the first five principal components. **(c) Variance explained.** Variance explained by each principal component.

## A.5 Relationship between interface contact density and interface quality scores

To assess how protein–protein interface contact density influences the stability of interface quality scores, we computed the variance of ICS, IPS, and QS_Best across the CASP15 and CASP16 community models for each target and examined their correlation with the native interface contact density. These datasets were selected because they consist of models from diverse sources for each target, reflecting the inherent variability in predicting structural models for a protein complex. Interface contact density was calculated as the ratio of residue-residue contacts to buried surface area (BSA). Residue pairs were considered in contact if any atoms were within 5.0 Å. BSA was computed using FreeSASA[42] with standard parameters as BSA(A,B) = SASA(chain A) + SASA(chain B) − SASA(A-B complex) for each chain pair, where SASA(chain) denotes the solvent-accessible surface area of the isolated chain, and SASA(A–B complex) represents that of the assembled complex. This difference quantifies the surface area that becomes inaccessible to solvent upon binding, corresponding to the interface area buried between the two chains. The mean contact density across all chain pairs is then used to compute the correlation with the variance of interface scores.

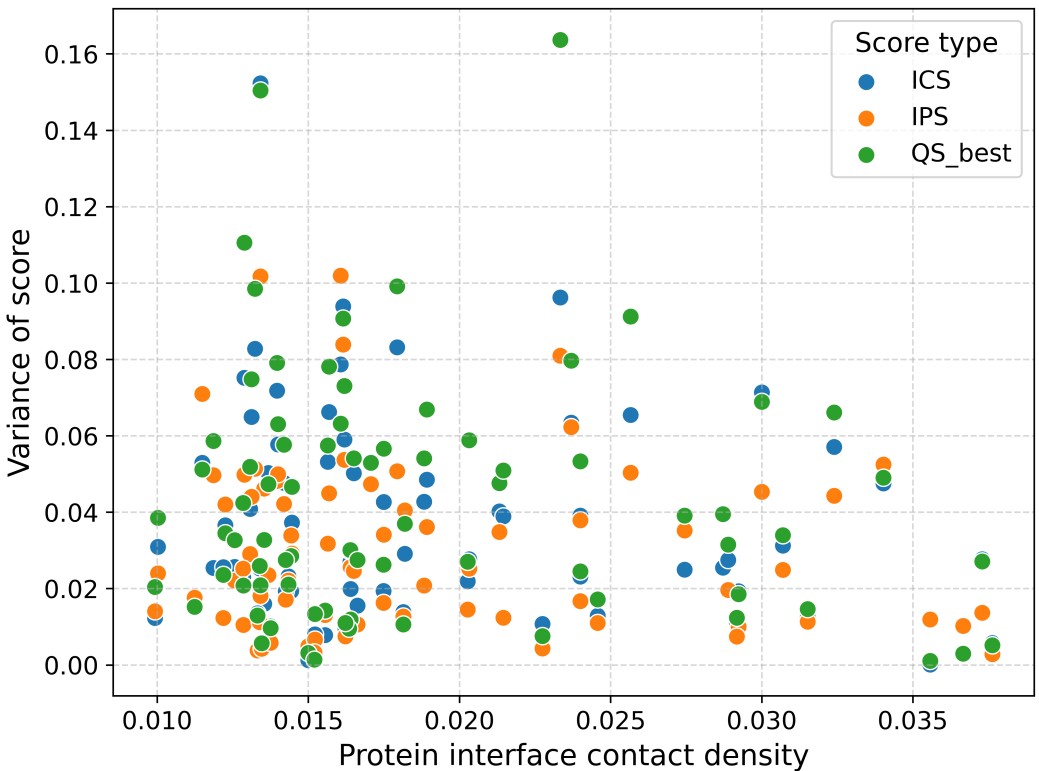

**Figure S3:** Variance of interface quality scores (ICS, IPS, QS_best) plotted against protein interface contact density. Weak negative correlations are observed for all three metrics ( r=-0.125 for ICS, r=-0.126 for IPS, and r=-0.115 for QS_best), indicating little influence of contact density on score variability.

The relationship between interface contact density and the variance in interface quality scores was examined (Fig. S3). All three metrics—ICS, IPS, and QS_Best—showed weak negative correlations with contact density, with correlation coefficients of $r = -0.125$ for ICS, $r = -0.126$ for IPS, and $r = -0.115$ for QS_Best. The results indicate a weak inverse relationship between interface contact density and interface score variability, suggesting that while contact density does have some influence on the variability of these interface quality scores, its effect is relatively modest.

## A.6 Guidance on Applications of Quality Scores

**Global quality scores.** TM-score variants (`tmscore`) are ideal for evaluating global fold accuracy in tasks such as structure-based function annotation and complex template retrieval, as they capture overall topology and are widely used in fold recognition benchmarks. RMSD is appropriate for atomic-level accuracy in applications such as structural refinement or drug binding site validation, where small positional deviations can impact function.

**Local quality scores.** Local Distance Difference Test (`lddt`) evaluates the accuracy of individual residue positions relative to their local environment, making it useful for residue-level analyses such as identifying flexible regions, validating side-chain placements, or assessing the accuracy of binding pocket geometry.

**Interface quality scores.** For interface-focused tasks such as docking, binder design, and epitope mapping, DockQ_Wave (`dockq_wave`) and Interface Patch Similarity (`ips`) provide comprehensive assessments of contact and interface quality. Interface Contact Precision (`ics_precision`) and Interface Contact Recall (`ics_recall`) evaluate the specificity and coverage of predicted contacts. The QS (global) score (`qs_global`) measures overall interface accuracy, while the QS (best) score (`qs_best`) highlights the highest-quality interface, which is particularly useful in asymmetric or functionally focused assemblies.

### A.7 Additional Input Features for Structural Models in CASP15_inhouse_dataset and CASP16_inhouse_dataset

Each structural model in the four datasets in PSBench is stored as a PDB file, which contains the (x, y, z) coordinates of every atom in the model. In addition, the two in-house datasets (CASP15_inhouse_dataset and CASP16_inhouse_dataset) and their subsets include the following extra features for each structural model, which can be leveraged by EMA methods.

- **model_type** : The type determines whether the model is generated using AlphaFold2 or AlphaFold3. AlphaFold3-based models are only available for CASP16_inhouse_dataset and its subset.

- **afm_confidence_score** : The AlphaFold2-Multimer Confidence score (`afm_confidence_score`) determines the confidence score in multimeric protein structures primarily assessed using ipTM and pTM score. For AlphaFold2 program, the AFM confidence score is available upon the completition of the prediction, but for AlphaFold3-based models, since the AFM confidence score is not readily available, it is obtained by the calculation $0.8 \times iptm + 0.2 \times ptm$.

- **af3_ranking_score** : The AlphaFold3 Ranking score (`af3_ranking_score`) determines the ranking score as provided by AlphaFold3 program. It is only available for AlphaFold3 generated models in CASP16_inhouse_dataset.

- **iptm** : The Interface Predicted Template Modeling score(`ipTM`) evaluates the accuracy of the predicted relative positioning of subunits within a protein-protein complex. Scores above 0.8 indicate confident, high-quality predictions, while scores below 0.6 typically reflect failed predictions. Values between 0.6 and 0.8 fall into an intermediate range, where prediction quality is uncertain and may vary.

- **num_inter_pae** : Number of inter-chain predicted aligned errors (<5 Å).

- **mpDockQ[43]/pDockQ[44]** : Multiple-interface predicted DockQ for multimer, or predicted DockQ (pDockQ) for dimer.

## A.8 CASP15_inhouse_dataset

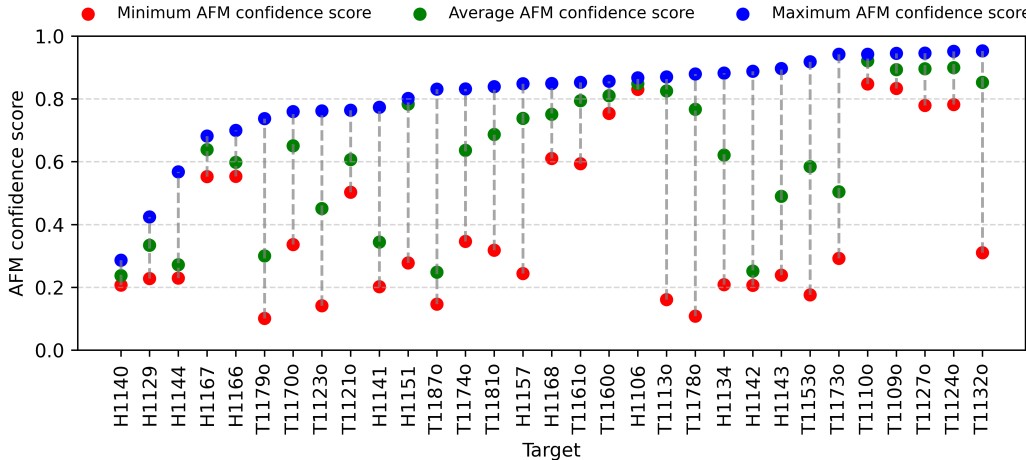

**Figure S4:** Distribution of AFM confidence scores per target in CASP15_inhouse_dataset.

**Table S1:** Summary of target information and number of models per target in CASP15_inhouse_dataset. It is worth noting that nine CASP15 targets (H1111, H1114, H1135, H1137, H1171, H1172, H1185, T1176o and T1192o) were excluded because they required alternative structure prediction approaches, such as template-based modeling, due to their large size or the limited number of the predicted structures (e.g., less than 30).

| Target | Stoichiometry | Protein Classification | Seq. Length | Total models |
|---|---|---|---|---|
| H1106 | A1B1 | CHAPERONE | 236 | 580 |
| H1129 | A1B1 | MEMBRANE PROTEIN | 1387 | 145 |
| H1134 | A1B1 | TOXIN/IMMUNE SYSTEM | 543 | 295 |
| H1140 | A1B1 | PROTEIN BINDING | 351 | 270 |
| H1141 | A1B1 | PROTEIN BINDING | 346 | 260 |
| H1142 | A1B1 | PROTEIN BINDING | 347 | 275 |
| H1143 | A1B1 | PROTEIN BINDING | 350 | 245 |
| H1144 | A1B1 | PROTEIN BINDING | 341 | 275 |
| H1151 | A1B1 | TRANSCRIPTION/Transferase | 228 | 265 |
| H1157 | A1B1 | OXIDOREDUCTASE | 1524 | 265 |
| H1166 | A1B1C1 | Unknown | 577 | 175 |
| H1167 | A1B1C1 | Unknown | 560 | 275 |
| H1168 | A1B1C1 | Unknown | 567 | 175 |
| T1109o | A2 | Unknown | 454 | 230 |
| T1110o | A2 | Unknown | 454 | 200 |
| T1113o | A2 | VIRAL PROTEIN | 386 | 415 |
| T1121o | A2 | DNA BINDING PROTEIN | 762 | 250 |
| T1123o | A2 | VIRAL PROTEIN | 532 | 215 |
| T1124o | A2 | TRANSFERASE | 768 | 200 |
| T1127o | A2 | Unknown | 422 | 235 |
| T1132o | A6 | CYTOSOLIC PROTEIN | 612 | 230 |
| T1153o | A2 | Unknown | 598 | 285 |
| T1160o | A2 | DNA BINDING PROTEIN | 96 | 275 |
| T1161o | A2 | DNA BINDING PROTEIN | 96 | 275 |
| T1170o | A6 | HYDROLASE | 1908 | 97 |
| T1173o | A3 | CELL ADHESION | 612 | 275 |
| T1174o | A3 | CELL ADHESION | 1014 | 215 |
| T1178o | A2 | VIRAL PROTEIN | 612 | 275 |
| T1179o | A2 | VIRAL PROTEIN | 522 | 275 |
| T1181o | A3 | Unknown | 2064 | 163 |
| T1187o | A2 | SUGAR BINDING PROTEIN | 332 | 275 |

**Table S2:** Model quality distribution in terms of dockq_wave thresholds (bad: score < 0.23, acceptable: 0.23 <= score <0.49, good: 0.49 <= score) for CASP15_inhouse_dataset.

| target | Number of bad models | Number of acceptable models | Number of good models |
|---|---|---|---|
| H1106 | 0 | 0 | 580 |
| H1129 | 145 | 0 | 0 |
| H1134 | 48 | 231 | 16 |
| H1140 | 269 | 1 | 0 |
| H1141 | 260 | 0 | 0 |
| H1142 | 275 | 0 | 0 |
| H1143 | 112 | 53 | 80 |
| H1144 | 260 | 15 | 0 |
| H1151 | 4 | 0 | 261 |
| H1157 | 58 | 202 | 5 |
| H1166 | 0 | 174 | 1 |
| H1167 | 0 | 215 | 60 |
| H1168 | 0 | 8 | 167 |
| T1109o | 1 | 189 | 40 |
| T1110o | 0 | 1 | 199 |
| T1113o | 9 | 0 | 406 |
| T1121o | 248 | 2 | 0 |
| T1123o | 95 | 118 | 2 |
| T1124o | 0 | 53 | 147 |
| T1127o | 4 | 12 | 219 |
| T1132o | 230 | 0 | 0 |
| T1153o | 98 | 2 | 185 |
| T1160o | 275 | 0 | 0 |
| T1161o | 275 | 0 | 0 |
| T1170o | 1 | 25 | 71 |
| T1173o | 181 | 67 | 27 |
| T1174o | 54 | 160 | 1 |
| T1178o | 25 | 116 | 134 |
| T1179o | 149 | 126 | 0 |
| T1181o | 162 | 1 | 0 |
| T1187o | 270 | 4 | 1 |

## A.9 CASP15_community_dataset

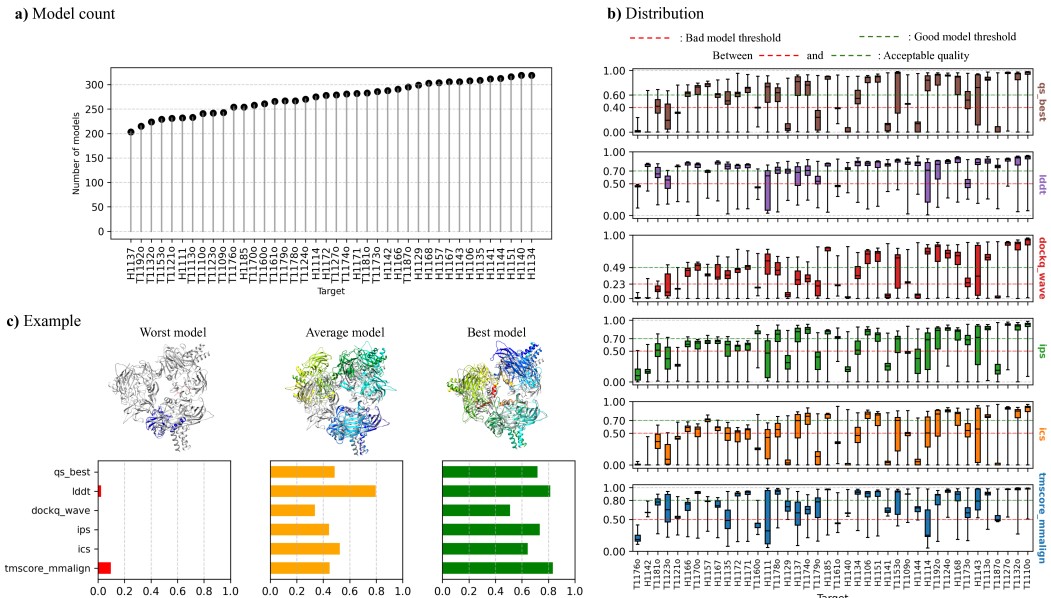

**Figure S5: CASP15_community_dataset. (a) Model count.** Number of models per target in the dataset. **(b) Score Distribution.** Box plots of each of six representative quality scores of the models for each target. **(c) Example.** Three representative models (worst, average, best) in terms of sum of the six representative quality scores for a target H1135. Each model with individual chains colored is superimposed with the true structure in gray.

**Table S3:** Summary of target information and the number of models per target in CASP15_community_dataset.

| Target | Stoichiometry | Protein Classification | Seq. Length | Total models |
|--------|---------------|------------------------|-------------|--------------|
| H1106 | A1B1 | CHAPERONE | 236 | 308 |
| H1111 | A9B9C9 | PROTEIN TRANSPORT | 8460 | 232 |
| H1114 | A4B8C8 | OXIDOREDUCTASE | 7988 | 275 |
| H1129 | A1B1 | MEMBRANE PROTEIN | 1387 | 299 |
| H1134 | A1B1 | TOXIN/IMMUNE SYSTEM | 543 | 319 |
| H1135 | A9B3 | STRUCTURAL PROTEIN | 1830 | 309 |
| H1137 | A1B1C1D1E1F1G2H1I1 | MEMBRANE PROTEIN | 4592 | 203 |
| H1140 | A1B1 | PROTEIN BINDING | 351 | 319 |
| H1141 | A1B1 | PROTEIN BINDING | 346 | 312 |
| H1142 | A1B1 | PROTEIN BINDING | 347 | 288 |
| H1143 | A1B1 | PROTEIN BINDING | 350 | 306 |
| H1144 | A1B1 | PROTEIN BINDING | 341 | 313 |
| H1151 | A1B1 | TRANSCRIPTION/Transferase | 228 | 316 |
| H1157 | A1B1 | OXIDOREDUCTASE | 1524 | 304 |
| H1166 | A1B1C1 | Unknown | 577 | 291 |
| H1167 | A1B1C1 | Unknown | 560 | 306 |
| H1168 | A1B1C1 | Unknown | 567 | 303 |
| H1171 | A6B1 | HYDROLASE | 1956 | 282 |
| H1172 | A6B2 | HYDROLASE | 2004 | 278 |
| H1185 | A1B1C1D1 | DNA BINDING PROTEIN | 1334 | 254 |
| T1109o | A2 | Unknown | 454 | 243 |
| T1110o | A2 | Unknown | 454 | 241 |
| T1113o | A2 | VIRAL PROTEIN | 386 | 233 |
| T1121o | A2 | DNA BINDING PROTEIN | 762 | 231 |
| T1123o | A2 | VIRAL PROTEIN | 532 | 242 |
| T1124o | A2 | TRANSFERASE | 768 | 270 |
| T1127o | A2 | Unknown | 422 | 279 |
| T1132o | A6 | CYTOSOLIC PROTEIN | 612 | 224 |
| T1153o | A2 | Unknown | 598 | 229 |
| T1160o | A2 | DNA BINDING PROTEIN | 96 | 261 |
| T1161o | A2 | DNA BINDING PROTEIN | 96 | 266 |
| T1170o | A6 | HYDROLASE | 1908 | 258 |
| T1173o | A3 | CELL ADHESION | 612 | 286 |
| T1174o | A3 | CELL ADHESION | 1014 | 281 |
| T1176o | A8 | UNKNOWN FUNCTION | 1360 | 254 |
| T1178o | A2 | VIRAL PROTEIN | 612 | 267 |
| T1179o | A2 | VIRAL PROTEIN | 522 | 267 |
| T1181o | A3 | Unknown | 2064 | 283 |
| T1187o | A2 | SUGAR BINDING PROTEIN | 332 | 295 |
| T1192o | A10 | DNA BINDING PROTEIN | 4180 | 215 |

**Table S4:** Model quality distribution in terms of dockq_wave thresholds (bad: score < 0.23, acceptable: 0.23 <= score < 0.49, good: 0.49 <= score) for CASP15_community_dataset.

| Target | Number of bad models | Number of acceptable models | Number of good models |
|---|---|---|---|
| H1106 | 40 | 6 | 262 |
| H1111 | 11 | 72 | 146 |
| H1114 | 20 | 15 | 238 |
| H1129 | 286 | 3 | 10 |
| H1134 | 70 | 164 | 85 |
| H1135 | 40 | 247 | 22 |
| H1137 | 59 | 99 | 45 |
| H1140 | 311 | 6 | 2 |
| H1141 | 300 | 1 | 11 |
| H1142 | 287 | 1 | 0 |
| H1143 | 129 | 40 | 137 |
| H1144 | 292 | 12 | 9 |
| H1151 | 54 | 12 | 250 |
| H1157 | 48 | 222 | 34 |
| H1166 | 16 | 224 | 51 |
| H1167 | 19 | 201 | 86 |
| H1168 | 18 | 39 | 246 |
| H1171 | 24 | 89 | 169 |
| H1172 | 27 | 207 | 44 |
| H1185 | 16 | 11 | 227 |
| T1109o | 37 | 187 | 19 |
| T1110o | 10 | 14 | 217 |
| T1113o | 25 | 7 | 201 |
| T1121o | 217 | 14 | 0 |
| T1123o | 155 | 78 | 9 |
| T1124o | 15 | 18 | 237 |
| T1127o | 8 | 8 | 263 |
| T1132o | 9 | 7 | 208 |
| T1153o | 68 | 16 | 145 |
| T1160o | 260 | 0 | 1 |
| T1161o | 259 | 4 | 3 |
| T1170o | 27 | 61 | 170 |
| T1173o | 133 | 114 | 39 |
| T1174o | 44 | 234 | 3 |
| T1176o | 254 | 0 | 0 |
| T1178o | 41 | 106 | 120 |
| T1179o | 171 | 86 | 10 |
| T1181o | 258 | 25 | 0 |
| T1187o | 272 | 5 | 18 |
| T1192o | 25 | 16 | 174 |

## A.10 CASP16_inhouse_dataset

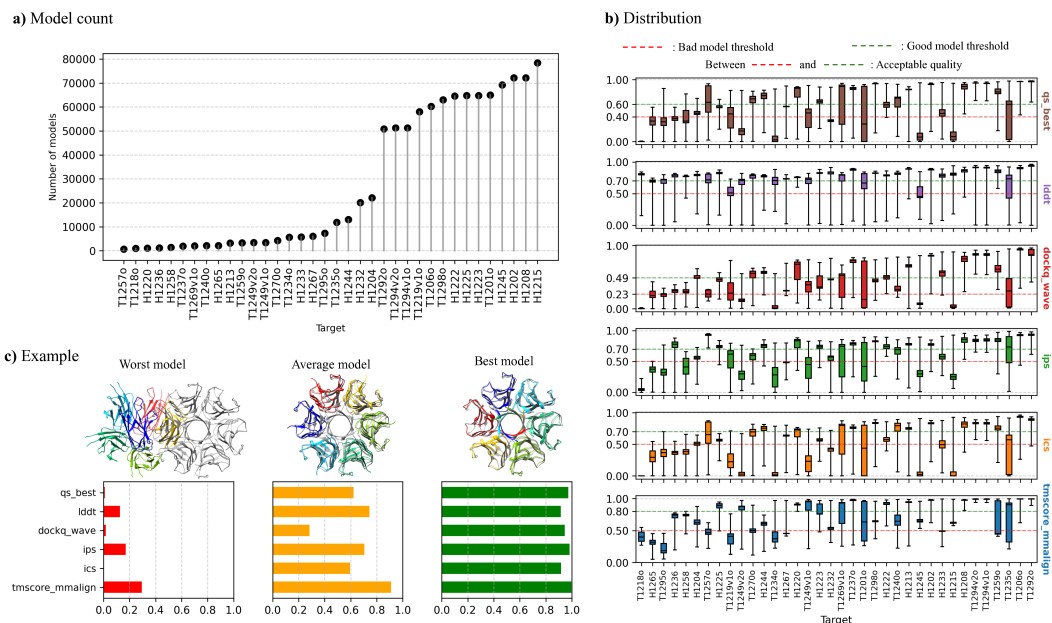

**Figure S6: CASP16_inhouse_dataset. (a) Model count.** Number of models per target in the dataset. **(b) Score Distribution.** Box plots of each of six representative quality scores of the models for each target. **(c) Example.** Three representative models (worst, average, best) in terms of sum of the six representative quality scores for a target T1235o. Each model with individual chains colored is superimposed with the true structure in gray.

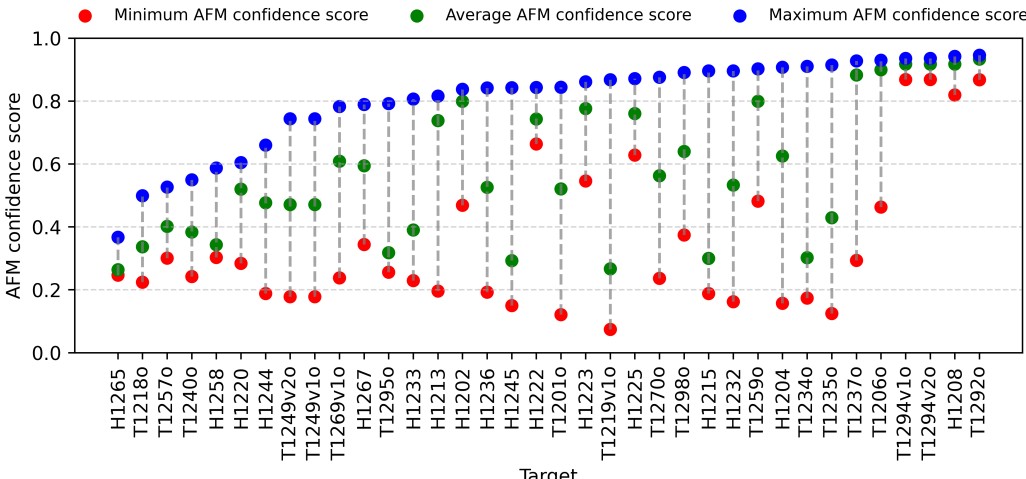

**Figure S7:** Distribution of AFM confidence scores of the structural models per target in CASP16_inhouse_dataset.

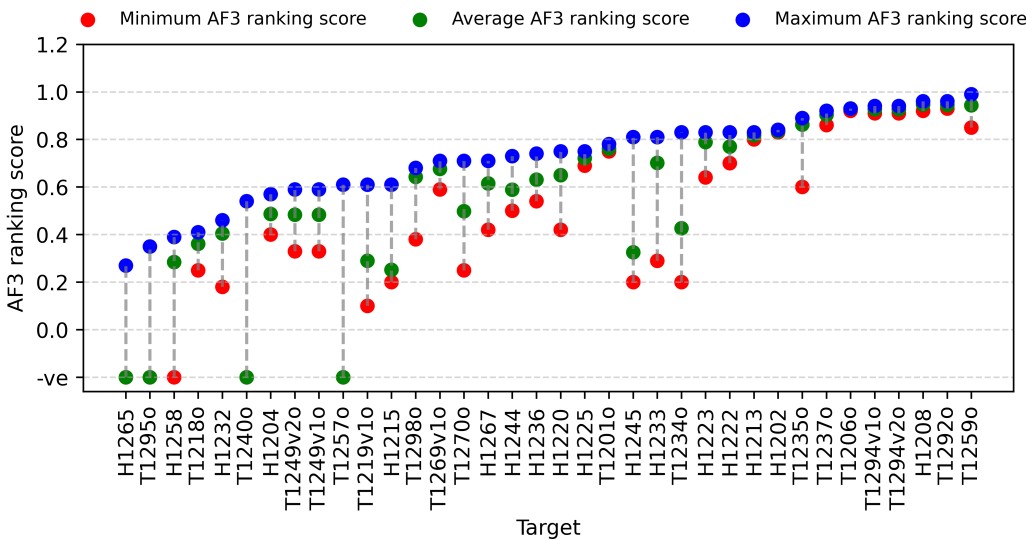

**Figure S8:** Distribution of AF3 ranking scores of the structural models per target in CASP16_inhouse_dataset.

**Table S5:** Summary of target information and number of models per target in CASP16_inhouse_dataset. It is worth noting that the length of three CASP16 targets (H1217, H1227, and H1272) exceeded the limit (about 5,000 residues) of running AlphaFold and required other structure prediction techniques such as template-based modeling. To make this dataset include only structural models generated by AlphaFold, they are excluded.

| Target | Stoichiometry | Protein Classification | Seq. Length | Total models |
|---|---|---|---|---|
| H1202 | A2B2 | SIGNALING PROTEIN | 380 | 72185 |
| H1204 | A2B2C2 | OXYGEN TRANSPORT | 858 | 22110 |
| H1208 | A1B1 | Unknown | 646 | 72200 |
| H1213 | A1B1C1D1E1 | Unknown | 1373 | 3200 |
| H1215 | A1B1 | Unknown | 369 | 78410 |
| H1220 | A1B4 | Unknown | 2515 | 1150 |
| H1222 | A1B1C1 | Unknown | 485 | 64600 |
| H1223 | A1B1C1 | Unknown | 486 | 64800 |
| H1225 | A1B1C1 | Unknown | 483 | 64799 |
| H1232 | A2B2 | VIRAL PROTEIN | 924 | 20090 |
| H1233 | A2B2C2 | VIRAL PROTEIN/IMMUNE SYSTEM | 1316 | 5700 |
| H1236 | A3B6 | VIRUS | 1929 | 1178 |
| H1244 | A2B2C2 | Unknown | 850 | 13000 |
| H1245 | A1B1 | Unknown | 317 | 69200 |
| H1258 | A1B2 | Unknown | 3092 | 1423 |
| H1265 | A9B18 | Unknown | 3924 | 2152 |
| H1267 | A2B2 | Unknown | 1852 | 6050 |
| T1201o | A2 | SIGNALING PROTEIN | 420 | 65020 |
| T1206o | A2 | VIRAL PROTEIN | 474 | 60205 |
| T1218o | A2 | Unknown | 2328 | 949 |
| T1219v1o | A10 | Unknown | 320 | 58000 |
| T1234o | A3 | VIRUS | 1239 | 5600 |
| T1235o | A6 | VIRUS | 690 | 11900 |
| T1237o | A4 | Unknown | 1952 | 1970 |
| T1240o | A3 | Unknown | 1959 | 2125 |
| T1249v1o | A3 | Unknown | 1464 | 3450 |
| T1249v2o | A3 | Unknown | 1464 | 3450 |
| T1257o | A3 | Unknown | 3789 | 712 |
| T1259o | A3 | Unknown | 729 | 3350 |
| T1269v1o | A2 | PROTEIN FIBRIL | 2820 | 2025 |
| T1270o | A6 | Unknown | 2622 | 4278 |
| T1292o | A2 | Unknown | 392 | 50800 |
| T1294v1o | A2 | Unknown | 428 | 51300 |
| T1294v2o | A2 | Unknown | 428 | 51300 |
| T1295o | A8 | Unknown | 3752 | 7369 |
| T1298o | A2 | Unknown | 684 | 63000 |

**Table S6:** Model quality distribution in terms of dockq_wave scores (bad: score < 0.23, acceptable: 0.23 <= score < 0.49, good: 0.49 <= score) for CASP16_inhouse_dataset.

| Target | Number of bad models | Number of acceptable models | Number of good models |
|---|---|---|---|
| H1202 | 0 | 448 | 71737 |
| H1204 | 144 | 12112 | 9854 |
| H1208 | 0 | 1135 | 71065 |
| H1213 | 100 | 9 | 3091 |
| H1215 | 77215 | 1024 | 171 |
| H1220 | 16 | 505 | 629 |
| H1222 | 0 | 25083 | 39517 |
| H1223 | 84 | 47480 | 17236 |
| H1225 | 20 | 46979 | 17800 |
| H1232 | 26 | 18138 | 1926 |
| H1233 | 2 | 883 | 4815 |
| H1236 | 258 | 920 | 0 |
| H1244 | 1 | 400 | 12599 |
| H1245 | 66839 | 1682 | 679 |
| H1258 | 252 | 1171 | 0 |
| H1265 | 1279 | 873 | 0 |
| H1267 | 1 | 6038 | 11 |
| T1201o | 34545 | 1227 | 29248 |
| T1206o | 0 | 1441 | 58764 |
| T1218o | 949 | 0 | 0 |
| T1219v1o | 25852 | 19207 | 12941 |
| T1234o | 5063 | 527 | 10 |
| T1235o | 5260 | 4097 | 2543 |
| T1237o | 2 | 29 | 1939 |
| T1240o | 80 | 2009 | 36 |
| T1249v1o | 803 | 2261 | 386 |
| T1249v2o | 3255 | 183 | 12 |
| T1257o | 352 | 360 | 0 |
| T1259o | 0 | 19 | 3331 |
| T1269v1o | 626 | 178 | 1221 |
| T1270o | 108 | 965 | 3205 |
| T1292o | 0 | 509 | 50291 |
| T1294v1o | 0 | 0 | 51300 |
| T1294v2o | 0 | 0 | 51300 |
| T1295o | 3668 | 3701 | 0 |
| T1298o | 1204 | 48835 | 12961 |

## A.11 CASP16_community_dataset

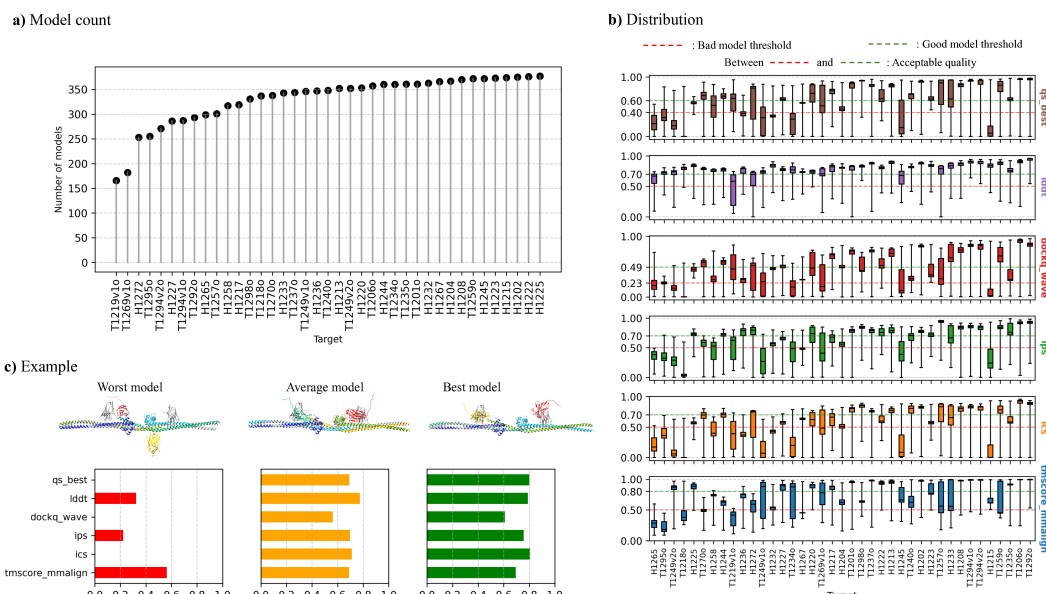

**Figure S9: CASP16_community_dataset. (a) Model count.** Number of models per target in the dataset. **(b) Score Distribution.** Box plots of each of six representative quality scores of the models for each target. **(c) Example.** Three representative models (worst, average, best) in terms of sum of the six representative quality scores for a target H1244. Each model with individual chains colored is superimposed with the true structure in gray.

**Table S7:** Summary of target information and number of models per target for CASP16_community_dataset.

| Target | Stoichiometry | Protein Classification | Seq. Length | Total models |
|---|---|---|---|---|
| H1202 | A2B2 | SIGNALING PROTEIN | 380 | 375 |
| H1204 | A2B2C2 | OXYGEN TRANSPORT | 858 | 367 |
| H1208 | A1B1 | Unknown | 646 | 370 |
| H1213 | A1B1C1D1E1 | Unknown | 1373 | 352 |
| H1215 | A1B1 | Unknown | 369 | 374 |
| H1217 | A2B2C2D2E2F2 | Unknown | 5878 | 319 |
| H1220 | A1B4 | Unknown | 2515 | 353 |
| H1222 | A1B1C1 | Unknown | 485 | 376 |
| H1223 | A1B1C1 | Unknown | 486 | 373 |
| H1225 | A1B1C1 | Unknown | 483 | 377 |
| H1227 | A1B6 | Unknown | 5689 | 286 |
| H1232 | A2B2 | VIRAL PROTEIN | 924 | 363 |
| H1233 | A2B2C2 | VIRAL PROTEIN/IMMUNE SYSTEM | 1316 | 343 |
| H1236 | A3B6 | VIRUS | 1929 | 347 |
| H1244 | A2B2C2 | Unknown | 850 | 360 |
| H1245 | A1B1 | Unknown | 317 | 372 |
| H1258 | A1B2 | Unknown | 3092 | 317 |
| H1265 | A9B18 | Unknown | 3924 | 299 |
| H1267 | A2B2 | Unknown | 1852 | 366 |
| H1272 | A1B1C1D1E1F1G1H1I1 | MEMBRANE PROTEIN | 6879 | 253 |
| T1201o | A2 | SIGNALING PROTEIN | 420 | 361 |
| T1206o | A2 | VIRAL PROTEIN | 474 | 357 |
| T1218o | A2 | Unknown | 2328 | 337 |
| T1219v1o | A10 | Unknown | 320 | 166 |
| T1234o | A3 | VIRUS | 1239 | 360 |
| T1235o | A6 | VIRUS | 690 | 361 |
| T1237o | A4 | Unknown | 1952 | 344 |
| T1240o | A3 | Unknown | 1959 | 348 |
| T1249v1o | A3 | Unknown | 1464 | 346 |
| T1249v2o | A3 | Unknown | 1464 | 352 |
| T1257o | A3 | Unknown | 3789 | 301 |
| T1259o | A3 | Unknown | 729 | 372 |
| T1269v1o | A2 | PROTEIN FIBRIL | 2820 | 182 |
| T1270o | A6 | Unknown | 2622 | 338 |
| T1292o | A2 | Unknown | 392 | 293 |
| T1294v1o | A2 | Unknown | 428 | 287 |
| T1294v2o | A2 | Unknown | 428 | 271 |
| T1295o | A8 | Unknown | 3752 | 255 |
| T1298o | A2 | Unknown | 684 | 331 |

**Table S8:** Model quality distribution based on dockq_wave scores (bad: score < 0.23, acceptable: 0.23 <= score < 0.49, good: 0.49 <= score) for CASP16_community_dataset.

| Target | Number of bad models | Number of acceptable models | Number of good models |
|---|---|---|---|
| H1202 | 2 | 37 | 336 |
| H1204 | 21 | 176 | 170 |
| H1208 | 31 | 15 | 324 |
| H1213 | 33 | 16 | 303 |
| H1215 | 333 | 12 | 29 |
| H1217 | 1 | 13 | 305 |
| H1220 | 13 | 174 | 166 |
| H1222 | 2 | 102 | 272 |
| H1223 | 0 | 239 | 134 |
| H1225 | 0 | 289 | 88 |
| H1227 | 5 | 86 | 195 |
| H1232 | 25 | 263 | 75 |
| H1233 | 12 | 35 | 296 |
| H1236 | 86 | 255 | 6 |
| H1244 | 39 | 15 | 306 |
| H1245 | 204 | 104 | 64 |
| H1258 | 53 | 262 | 2 |
| H1265 | 208 | 86 | 5 |
| H1267 | 31 | 331 | 4 |
| H1272 | 69 | 33 | 151 |
| T1201o | 64 | 8 | 289 |
| T1206o | 27 | 30 | 300 |
| T1218o | 286 | 43 | 8 |
| T1219v1o | 32 | 52 | 82 |
| T1234o | 194 | 162 | 4 |
| T1235o | 52 | 238 | 71 |
| T1237o | 16 | 19 | 309 |
| T1240o | 22 | 321 | 5 |
| T1249v1o | 170 | 160 | 16 |
| T1249v2o | 303 | 46 | 3 |
| T1257o | 85 | 112 | 104 |
| T1259o | 11 | 9 | 352 |
| T1269v1o | 94 | 28 | 60 |
| T1270o | 22 | 57 | 259 |
| T1292o | 4 | 11 | 278 |
| T1294v1o | 4 | 3 | 280 |
| T1294v2o | 12 | 3 | 256 |
| T1295o | 117 | 138 | 0 |
| T1298o | 26 | 194 | 111 |

## A.12 Multimer_7_2024_8_2025_dataset

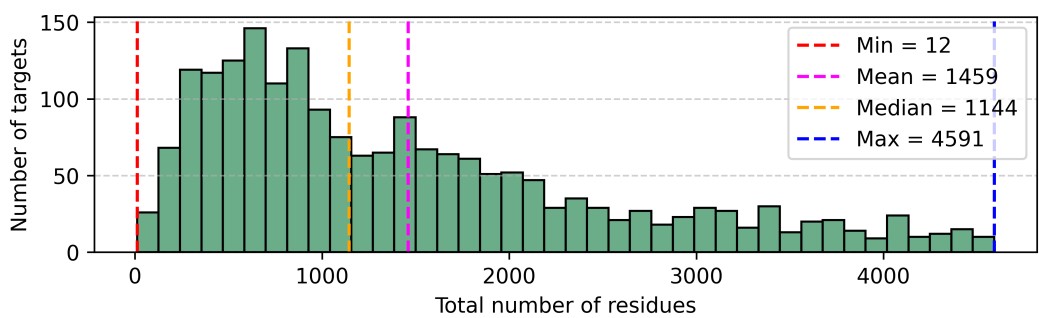

**Figure S10:** Distribution of residue count per target in Multimer_7_2024_8_2025_dataset.

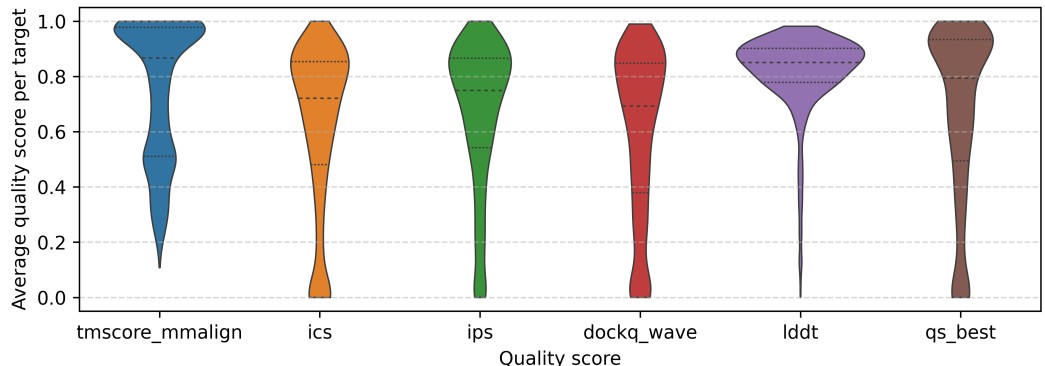

**Figure S11:** Violin plot of distribution of average quality scores of the structural models per target in Multimer_7_2024_8_2025_dataset.

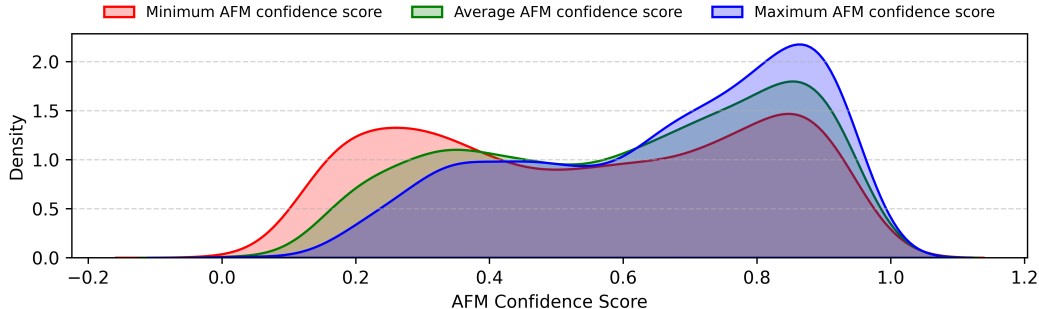

**Figure S12:** Density of AlphaFold-Multimer-style (AFM) confidence scores of the structural models per target in Multimer_7_2024_8_2025_dataset.

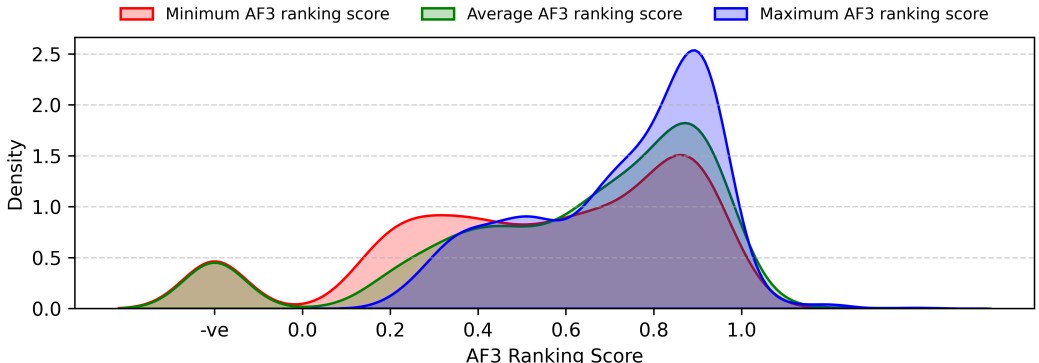

**Figure S13:** Density of AlphaFold3 (AF3) ranking scores of the structural models per target in Multimer_7_2024_8_2025_dataset.

### A.13 CASP15_inhouse_TOP5_dataset and CASP16_inhouse_TOP5_dataset

CASP15_inhouse_TOP5_dataset and CASP16_inhouse_TOP5_dataset are the subset of CASP15_inhouse_dataset and CASP16_inhouse_dataset respectively. Each contains only top 5 models for each target predicted by each of dozens of AlphaFold-based predictors in our MULTI-COM protein structure prediction system during CASP15 or CASP16, even though each predictor might generate many (e.g., hundreds of) models. These two subsets were used to train and evaluate GATE-AFM. The number of models per target in each of these two subsets is given in the Table S9.

**Table S9:** Summary of the CASP15 and CASP16 in-house TOP5 datasets.

| CASP15_inhouse_TOP5_dataset | | CASP16_inhouse_TOP5_dataset | |
|---|---|---|---|
| Target | Total Models | Target | Total Models |
| H1106 | 290 | H1202 | 390 |
| H1129 | 65 | H1204 | 350 |
| H1134 | 150 | H1208 | 380 |
| H1140 | 130 | H1213 | 300 |
| H1141 | 125 | H1215 | 355 |
| H1142 | 125 | H1220 | 101 |
| H1143 | 115 | H1222 | 345 |
| H1144 | 125 | H1223 | 345 |
| H1151 | 120 | H1225 | 345 |
| H1157 | 115 | H1232 | 250 |
| H1166 | 75 | H1233 | 255 |
| H1167 | 85 | H1236 | 100 |
| H1168 | 75 | H1244 | 330 |
| T1109o | 115 | H1245 | 360 |
| T1110o | 100 | H1258 | 90 |
| T1113o | 210 | H1265 | 77 |
| T1121o | 120 | H1267 | 385 |
| T1123o | 110 | T1201o | 345 |
| T1124o | 100 | T1206o | 325 |
| T1127o | 120 | T1218o | 35 |
| T1132o | 115 | T1219v1o | 305 |
| T1153o | 130 | T1234o | 325 |
| T1160o | 125 | T1235o | 295 |
| T1161o | 125 | T1237o | 275 |
| T1170o | 45 | T1240o | 155 |
| T1173o | 125 | T1257o | 73 |
| T1174o | 95 | T1259o | 305 |
| T1178o | 125 | T1269v1o | 185 |
| T1179o | 125 | T1270o | 290 |
| T1181o | 65 | T1292o | 270 |
| T1187o | 125 | T1295o | 230 |
| | | T1298o | 325 |

## A.14 Analysis of Potential Bias from AlphaFold-Dominated Model Generation

An issue for PSBench is the potential bias introduced by the dominance of AlphaFold in model generation. Although the two community datasets (CASP15_community_dataset and CASP16_community_dataset) contain models from a variety of predictors, the majority of their models still originated from AlphaFold. The other three datasets in PSBench consist of AlphaFold-generated models only. This reflects the current landscape of protein structure prediction but may bias benchmark outcomes toward AlphaFold-specific characteristics.

To assess this bias, we compared TM-score distributions between the CASP community datasets (23,841 models from AlphaFold and non-AlphaFold predictors) and the AlphaFold-only in-house datasets. Community datasets exhibited a lower average TM-score (0.7166 vs. 0.7873) and a higher standard deviation (0.2394 vs. 0.2160), indicating slightly reduced average quality and greater variability compared to AlphaFold-only models.

We further examined the impact of this bias on EMA performance. Specifically, we compared GATE (trained on CASP15_community_dataset) and GATE-AFM (trained on CASP15_inhouse_TOP5_dataset) on the CASP16_inhouse_TOP5_dataset, which represents high-confidence AlphaFold predictions. As shown in Table S10 GATE-AFM outperformed GATE in

Spearman's correlation, loss, and AUROC for both TM-score and DockQ_Wave, while GATE achieved slightly higher Pearson's correlation on TM-score.

These results suggest that the training dataset composition can affect EMA generalization performance. We therefore caution users to consider model-generation bias when interpreting benchmark outcomes. We plan to incorporate models from emerging prediction methods in future PSBench releases to mitigate this bias.

**Table S10:** Performance of GATE-AFM and GATE on the CASP16_inhouse_TOP5_dataset. Metrics include Pearson's correlation (Corr$^P$), Spearman's correlation (Corr$^S$), ranking loss, and AUROC (75$^{th}$ percentile cutoff) reported separately for TM-score and DockQ_wave. Bold values indicate better performance.

| Method | TM-score | | | | DockQ_wave | | | |
|---|---|---|---|---|---|---|---|---|
| | Corr$^P$ ↑ | Corr$^S$ ↑ | Loss ↓ | AUROC ↑ | Corr$^P$ ↑ | Corr$^S$ ↑ | Loss ↓ | AUROC ↑ |
| GATE-AFM | 0.372 | **0.283** | **0.102** | **0.658** | **0.431** | **0.322** | **0.138** | **0.662** |
| GATE | **0.408** | 0.277 | 0.133 | 0.647 | 0.380 | 0.300 | 0.163 | 0.648 |

# B   Experimental Design

## B.1   Standard EMA Methods in PSBench

PSBench includes six standard EMA methods that are publicly available. They serve as baseline methods for comparison with new EMA methods. Below is a brief overview of each method and its availability:

- **AlphaFold2-Multimer Confidence score (AFM Confidence)**[10]: AlphaFold2-Multimer provides self-estimated accuracy estimates for its predicted structures using a confidence score that is computed as a weighted sum of ipTM (interface predicted TM-score) and pTM (predicted TM-score), specifically: 0.8 * ipTM + 0.2 * pTM. This score serves as a single-model EMA method and a strong baseline for datasets generated by AlphaFold2-Multimer or AlphaFold3.

- **GATE**[18]:  A multi-model EMA approach that leverages graph transformers applied to pairwise similarity graphs derived from input models. GATE combines both single-model and multi-model quality scores from individual models with comparative geometric similarities between models, enabling it to effectively predict the global structural accuracy (e.g., TM-score) of complex structural models. Source code is available at: `https://github.com/BioinfoMachineLearning/gate`. **GATE-AFM**: An enhanced variant of GATE by using AlphaFold2-Multimer features as additional node features. It can be used if such features are available.

- **DProQA**[31]: A single-model EMA method based on a Gated Graph Transformer architecture that modulates local neighborhood interactions. It is specifically designed to estimate the interface quality of protein complex models (e.g., DockQ scores) by leveraging a K-nearest neighbor (K-NN) graph representation of the complex structure. Source code is available at: `https://github.com/jianlin-cheng/DProQA`.

- **VoroMQA-dark, VoroIF-GNN-score, VoroIF-GNN-pCAD-score**[38]: A set of single-model EMA methods that utilize the VoroIF-GNN framework (Voronoi Interface Graph Neural Network) to assess protein complex interface quality. These methods operate on Voronoi tessellation-based atomic contact areas, capturing geometric and topological features of the interface. Source code is available at: `https://github.com/kliment-olechnovic/ftdmp`.

- **GCPNet-EMA**[39]: An EMA extension of GCPNet (Geometry-Complete Perceptron Network), a deep graph neural network that constructs a 3D graph representation from the atomic point cloud of a protein structure. It predicts both per-residue and per-model structural accuracy estimates, such as local and global lDDT. Although GCPNet-EMA is

originally trained on tertiary protein structures (e.g., single-chain models), it can be directly applied to evaluate the accuracy of protein complex structures. Source code is available at: `https://github.com/BioinfoMachineLearning/GCPNet-EMA`.

- **Average Pairwise Similarity Score (PSS)**[40]: A multi-model EMA method that evaluates each predicted complex by computing the average pairwise TM-score between it and all other models in the structural pool using MMalign. This simple yet effective consensus-based approach serves as a strong baseline for estimating the quality of the protein complex model. Source code is available at: `https://github.com/BioinfoMachineLearning/MULTICOM_qa`, with a simplified implementation available at: `https://github.com/BioinfoMachineLearning/gate/blob/main/gate/feature/mmalign_pairwise.py`.

It is worth noting that during benchmarking, the quality scores predicted by DProQA, VoroMQA and VoroIF-GNN, and GCPNet-EMA were normalized by multiplying the raw score by the ratio of the model length to the native structure length. This normalization penalizes shorter decoys, ensuring that the scores more accurately reflect both the completeness and the accuracy of the predicted models relative to the native structures.

## B.2   The Details of Training and Validating GATE

GATE and GATE-AFM were first trained, validated, and tested on the CASP15 datasets (i.e., CASP15_community_dataset or CASP15_inhouse_dataset) respectively and then were blindly evaluated on unseen targets in the CASP16 datasets (i.e., CASP16_community_dataset or CASP16_inhouse_dataset) during the CASP16 competition from May to August, 2024.

**Graph construction and architecture.** To predict the quality scores for a set of complex structural models of a protein, GATE and GATE-AFM construct a pairwise model similarity graph in which each node represents a model, and an edge is formed between two nodes if the TM-score between the corresponding models exceeds 0.5. Each node is annotated with both single-model quality scores (e.g., ICPS, EnQA, DProQA, VoroMQA) and aggregated pairwise similarity scores (e.g., TM-score, QS-score, DockQ, CAD-score). For GATE-AFM, additional AlphaFold2-Multimer–specific features (confidence scores, ipTM, inter-chain predicted alignment errors, mpDockQ) are incorporated as node features. Edge features encode the pairwise similarity scores between connected models, including TM-score, QS-score, DockQ, and CAD-score.

From each full graph, 2,000–3,000 subgraphs are sampled, each containing up to 50 nodes. Within a subgraph, node and edge features are embedded, updated through graph transformer layers with multi-head attention and feed-forward networks, and passed through a multilayer perceptron (MLP) to predict a quality score for each node. The models are trained with a weighted loss combining a pointwise mean squared error (MSE) term (predicted vs. true scores) and a pairwise loss term (mean absolute error between predicted and true differences of model pairs). The pointwise loss weight was fixed at 1, while the pairwise loss weight was tuned as a hyperparameter.

**Training protocol.** GATE was trained and validated on the CASP15_community_dataset, which comprises 10,935 models of 40 protein targets, plus 187 models from another target (T1115o) whose native structure is not publicly available. For the 10,935 models, usalign_tmscore was used as labels. For T1115o, TM-scores were obtained from the CASP15 website. CASP15_community_dataset was partitioned into training, validation, and test sets using 10-fold cross-validation split by targets. For each target, 2,000 subgraphs were sampled, leading to 8,000–10,000 subgraphs per subset. Eight folds were used for training, one for validation, and one for testing, iterating across all folds. The fold assignments are listed in Table S11. Hyperparameter search space is shown in Table S12.

**GATE-AFM.** GATE-AFM was trained using the same cross-validation protocol and graph construction process as GATE, but on the CASP15_inhouse_TOP5_dataset (31 protein complex targets). To augment the data, 3,000 subgraphs per target were sampled. The only difference from GATE is that GATE-AFM incorporates AlphaFold2-Multimer–specific features as additional node features. TM-score labels were generated with an older version of the in-house TM-score script (tmscore_usalign_aligned_v0), which was later updated in CASP16 to correct minor alignment issues.

The updated version is included in PSBench, but the original scores remain available in the dataset to support reproducibility. The fold assignments are listed in Table S13.

**Table S11:** Targets assigned to each fold in the CASP15_community_dataset for training GATE.

| Fold | Targets |
|------|---------|
| Fold0 | H1135, T1127o, T1161o, T1132o, H1144 |
| Fold1 | H1151, T1153o, H1171, H1114 |
| Fold2 | T1170o, H1166, T1176o, H1134 |
| Fold3 | H1111, H1106, T1109o, T1121o |
| Fold4 | T1174o, T1115o, H1172, H1143 |
| Fold5 | H1137, H1142, T1192o, H1140 |
| Fold6 | T1187o, T1181o, T1179o, T1178o |
| Fold7 | H1168, T1173o, T1160o, H1167 |
| Fold8 | T1113o, H1185, T1123o, H1157 |
| Fold9 | T1110o, H1141, T1124o, H1129 |

**Table S12:** Hyperparameter search space explored during model fine-tuning.

| Hyperparameter | Candidate Values |
|----------------|------------------|
| Number of attention heads | 4, 8 |
| Number of graph transformer layers | 2, 3, 4, 5 |
| Dropout rate | 0.1, 0.2, 0.3, 0.4, 0.5 |
| MLP dropout rate | 0.1, 0.2, 0.3, 0.4, 0.5 |
| Hidden dimension | 16, 32, 64 |
| Weight of the pairwise MSE loss | `auto`, 0.01, 0.05, 0.1, 0.2, 0.3, 0.4, 0.5, 0.6, 0.7, 0.8, 0.9, 1 |
| Optimizer | `AdamW`, `SGD` |
| Learning rate | 1e-5, 5e-5, 1e-4, 5e-4, 1e-3 |
| Weight decay | 0.01, 0.05 |
| Layer normalization | False, True |
| Batch size | 256, 400, 512 |

**Table S13:** Targets assigned to each fold in the CASP15_inhouse_dataset for training GATE-AFM.

| Fold | Targets |
|------|---------|
| Fold0 | T1127o, T1161o, T1132o, T1174o |
| Fold1 | T1160o, H1106, H1134 |
| Fold2 | T1173o, T1178o, T1110o |
| Fold3 | T1170o, H1142, H1140 |
| Fold4 | T1179o, T1187o, T1153o |
| Fold5 | T1123o, H1151, T1181o |
| Fold6 | T1121o, T1113o, H1144 |
| Fold7 | H1167, T1124o, H1157 |
| Fold8 | H1168, H1143, H1166 |
| Fold9 | H1141, T1109o, H1129 |

## B.3 Computational Requirements for Training on PSBench

The computational cost of training EMA methods on the full PSBench dataset depends heavily on how structural models are used as inputs. For single-model EMA methods, where each model is processed independently, training time scales linearly with the number of models (e.g., 10,942 models in the CASP15_community_dataset). In contrast, multi-model methods such as GATE construct a pairwise

model similarity graph for each target using all available models, which introduces quadratic or higher time complexity with respect to the number of models per target. However, once the similarity graph is constructed, extracting subgraphs and using them for training is considerably faster. Under this setup, GATE was trained using a single NVIDIA A100 GPU (80 GB) and required approximately 2 minutes per epoch. Researchers with smaller GPU memory may still train on PSBench by applying subgraph sampling or filtering strategies (e.g., using top-ranked models per target).

## B.4 Threshold Selection for AUROC Evaluation

In the main experiments, we used the 75th percentile of the ground-truth scores (i.e., CASP official scores) for each target to define "high-quality" models in a relative sense. This percentile-based threshold adapts to the difficulty of individual targets and ensures balanced positive/negative class distribution for AUROC calculations across targets.

To assess the impact of threshold choice, we additionally evaluated AUROC using fixed cutoffs that are commonly used in the field. Specifically, we applied TM-score $\geq 0.5$ and DockQ_Wave $\geq 0.49$ to define high-quality models. Table S14 presents the performance of the EMA methods in PSBench under both percentile-based and fixed-threshold definitions on the CASP16_inhouse_TOP5_dataset. Under the fixed thresholds, GATE-AFM achieved AUROC scores of 0.635 (TM-score) and 0.684 (DockQ_Wave), which is ranked second among all the methods.

Statistical significance was assessed using a one-sided Wilcoxon signed-rank test comparing GATE-AFM with each baseline method. As indicated in Table S14, results marked with an asterisk (*) denote significant differences ($p < 0.05$). Under the fixed-threshold setting, GATE-AFM maintained strong performance and significantly outperformed most baseline methods, including AFM-Confidence, VoroIF-GNN, VoroIF-GNN-pCAD, and DProQA, across both TM-score and DockQ_Wave metrics. These results demonstrate that GATE-AFM remains one of the top-performing and statistically robust EMA methods, with consistent relative ranking and performance patterns regardless of the thresholding strategy.

**Table S14:** AUROC values of different EMA methods under 75th percentile and fixed threshold (0.5 and 0.49) definitions for TM-score and DockQ_Wave on CASP16_inhouse_TOP5_dataset. Significant difference ($p < 0.05$) between GATE-AFM and other methods based on the one-sided Wilcoxon signed-rank test is marked with *.

| Methods | TM-score | | DockQ_Wave | |
|---|---|---|---|---|
| | 75th pct. | 0.5 cutoff | 75th pct. | 0.49 cutoff |
| GATE-AFM | **0.658** | 0.635 | **0.662** | 0.684 |
| AFM-Confidence | 0.597* | 0.585* | 0.593* | 0.673 |
| PSS | 0.647 | **0.650** | 0.645 | **0.685** |
| GCPNet-EMA | 0.643 | 0.606* | 0.648 | 0.672 |
| VoroMQA-dark | 0.609 | 0.570* | 0.622 | 0.588* |
| VoroIF-GNN-pCAD | 0.589* | 0.541* | 0.615 | 0.570* |
| VoroIF-GNN | 0.599* | 0.565* | 0.622 | 0.601* |
| DProQA | 0.569* | 0.543* | 0.587* | 0.618 |

## B.5 Failure Mode Analysis

To better understand the limitations of EMA methods, we examined failed cases (targets) with large prediction errors from GATE-AFM. Two recurring failure modes were identified.

For targets like H1202 and T1206o, EMA methods show high MAE (mean average error) despite the predicted models being generally good and similar. In these cases, GATE-AFM preserves relative rankings (low ranking loss) but underestimates absolute TM-scores, indicating a calibration issue. Its focus on pairwise relationships can compress score predictions, reducing absolute accuracy.

In contrast, H1265 includes only low-quality, diverse models with no resemblance to the native structure. Here, structural similarity among poor models misleads GATE-AFM, causing TM-score overestimation due to an uninformative similarity graph.

These cases reveal two distinct failure modes: score compression in high-quality model sets and misleading graph structures in poor-quality sets. By comparison, targets like H1215 and H1223, with varied and well-formed models, show low MAE and strong performance.

## B.6  Ranking of EMA predictors on the CASP16_community_dataset based on Z-scores

To assess the overall performance of EMA predictors on the CASP16_community_dataset, we ranked the 38 EMA predictors based on cumulative positive Z-scores. For each target, the Z-scores for the predictors were computed separately for the four evaluation metrics (Pearson's correlation, Spearman's correlation, ranking loss, and AUROC) based on TM-score. The z-score for a predictor for a target is equal to the original score minus the average score of all the predictors divided by the standard deviation. When calculating total Z-scores for each predictor, only positive Z-scores for each target were accumulated to emphasize strong performances.

In this ranking, MULTICOM_GATE achieved third place with a total Z-score of 95.7, closely following ModFOLDdock2 (96.9) and MULTICOM_LLM (105.4).

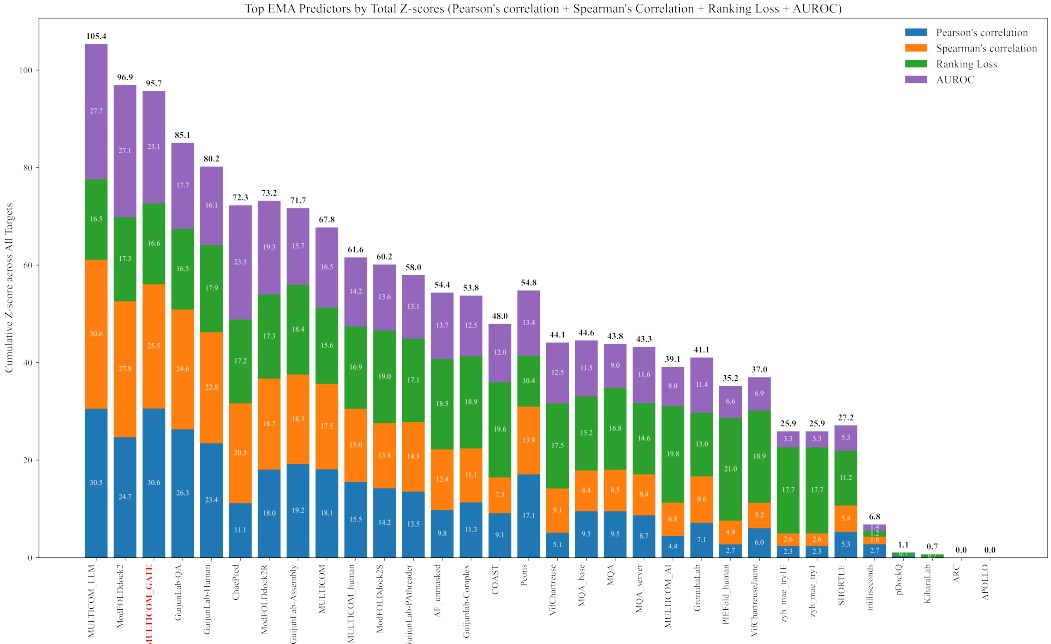

**Figure S14: CASP16_community_dataset results.** Stacked bar plot showing the cumulative positive Z-scores across all CASP16 community targets for the participated EMA predictors. Contributions from four performance metrics (Pearson's correlation, Spearman's correlation, ranking loss, and AUROC) are stacked to highlight the overall performance. EMA predictors are ranked by their total Z-scores. MULTICOM_GATE is colored in red.

## B.7  Runtime and Memory Usage of EMA Methods

We compared the runtime and memory usage of representative EMA methods in PSBench on the computer system described in Appendix C.1. For GATE-AFM and GATE, the inference time after generating quality features for each model using external EMA methods is reported. Information for AFM-Confidence is not included since its estimates are generated alongside AlphaFold model predictions and are not obtained through a separate inference step.

Table S15 summarizes the results. Multi-model EMA methods such as GATE, GATE-AFM, and PSS generally provide higher predictive accuracy but require more memory and longer runtimes. In contrast, single-model methods such as GCPNet-EMA and DProQA are more lightweight but typically achieve lower accuracy. VoroMQAs were run on CPU in our experiments and therefore required longer runtimes compared to GCPNet-EMA or DProQA.

**Table S15:** Peak memory usage and runtime of EMA methods in PSBench.

| EMA Method | CPU Mem (GB) | GPU Mem (GB) | Runtime (min) |
|---|---|---|---|
| GATE-AFM (inference) | 100.9 | 11.9 | 12.98 |
| GATE (inference) | 100.1 | 12.0 | 31.04 |
| PSS | 0.53 | N/A | 18.88 |
| GCPNet-EMA | 0.53 | 0.64 | 11.13 |
| VoroMQAs (CPU) | 0.53 | N/A | 19.28 |
| DProQA | 0.53 | 2.63 | 4.72 |

## C  System Requirements

### C.1  Environment for benchmarking EMA methods with PSBench and labeling model datasets

The generation of model quality scores and the evaluation of baseline EMA methods were performed on a computing server with the following specifications:

- **Operating System:** CentOS Linux
- **CPU:** AMD EPYC 7552, 3.2 GHz, 48 cores
- **RAM:** 50 GB
- **GPU:** NVIDIA A100, 80 GB (not required for quality score annotation, only needed to run some EMA methods)

### C.2  Environment for training and validating GATE

The training and validation of the GATE and GATE-AFM were conducted on a high-performance computing system with the following configuration:

- **Operating System:** CentOS Linux
- **CPU:** AMD EPYC 7552, 3.2 GHz, 48 cores
- **RAM:** 500 GB
- **GPU:** NVIDIA A100, 80 GB

