# OpenReview forum: "PSBench: a large-scale benchmark for estimating the accuracy of protein complex structural models"
_NeurIPS.cc/2025/Datasets_and_Benchmarks_Track — NeurIPS 2025 Datasets and Benchmarks Track poster_

### Official Review · Reviewer_DYzw · 2025-07-01

**Ethics Flags:** Data privacy, copyright, and consent
**Rating:** 4
**Confidence:** 5

**Summary:**

To address the lack of large, diverse, and well-annotated datasets for training and evaluation of machine learning-based EMA methods during protein prediction, the paper propose PSBench, a benchmark suite. The PSBench provides more than one million complex structural models generated by the SOTA methods. The structural models were generated for 79 diverse protein complex targets, and assigned 10 complementary quality scores at the local, glocal, and interface levels. The PSBench provieds automated evaluation tools for comparing new EMA methods with 6 standard baseline EMA methods, and was rigorously proved in CASP16.

**Dataset Code Accessibility:**

Yes

**Ethical Considerations:**

No, there are no or only very minor ethics concerns

**Limitations Weaknesses:**

1. The effectiveness of the new method demonstrated in the PSBench is too limited.

**Strengths Contributions:**

1. The paper's presentation is well-written, organized, and easy to understand.
2. The benchmark suite (PSBench) proposed by the paper makes the unique contributions to the field of protein prediction. The PSBench own more than one million complex structural models. The structural models were generated for 79 diverse protein complex targets, and a wide  range of sequence lengths. The PSBench provides an automated evaluation tools for comparing new EMA methods with 6 standard baseline EMA methods.

---

> ### Author Rebuttal · Authors · 2025-07-30
>
> Thank you very much for the very constructive and insightful assessment of the strengths and limitations of our manuscript.
>
> > **Comment 1: The effectiveness of the new method demonstrated in the PSBench is too limited.**
>
> We appreciate your concern regarding the demonstration of the effectiveness of the new EMA methods (GATE and GATE-AFM) using PSBench.
>
> The primary goal of this work is to introduce PSBench, a reproducible and extensible benchmark designed to support rigorous development and evaluation of EMA methods and demonstrate its effectiveness. As you pointed out, the benchmark includes four datasets comprising more than one million protein complex structural models, standardized quality annotations, a diverse set of ten established evaluation metrics, and multiple baseline EMA methods—forming the foundation for systematic EMA method development and comparison.
>
> Using only a portion of the PSBench datasets (CASP15_community_dataset and CASP15_inhouse_dataset), we trained and evaluated GATE and GATE-AFM, two new graph transformer-based EMA methods for predicting the accuracy of protein complex structural models in early 2024. We then blindly tested them in the 16th Critical Assessment of Techniques for Protein Structure Prediction (CASP16) from May to August 2024.
>
> As shown in the manuscript (Section 5.1), GATE-AFM achieved superior performance in ranking loss and AUROC compared to multiple highly effective EMA baselines—including AFM-Confidence (AlphaFold confidence score), PSS, and VoroMQA/VoroIF variants—across both TM-score and interface-focused metrics like DockQ on the CASP16 datasets according to our assessment. Some improvement is statistically significant.
>
> Moreover, according to the official CASP16 assessment, GATE was ranked among the top CASP16 EMA methods out of 38 CASP16 predictors in estimating the accuracy of the CASP16 community models. Due to the outstanding performance, our group was invited by CASP organizers to present it at the CASP16 meeting held in December 2024 and to contribute a description of our CASP16 EMA predictors to the joint CASP16 EMA assessment paper [1], led by the CASP16 EMA assessors. Only five top EMA prediction groups in the world received this recognition. The official CASP16 EMA assessment paper has been recently accepted to be published in the CASP16 special issue of Proteins journal, further confirming the effectiveness of GATE. The success of GATE in the world-wide CASP16 competition rigorously demonstrates that the two datasets (CASP15_community_dataset and CASP15_inhouse_dataset) are highly effective for developing and evaluating protein complex EMA methods. With the addition of the newer, larger CASP16_community_dataset and CASP16_inhouse_dataset, we are confident that PSBench will be even more effective in this regard.
>
> In the final version of this manuscript, we will refer readers to the official CASP16 assessment paper [1] for more details about the performance of GATE and its effectiveness that has been blindly and comprehensively evaluated by the external CASP EMA expert assessors.  This addition should greatly strengthen the demonstration of the effectiveness of our EMA method trained with PSBench.
>
> In summary, we believe that the effectiveness of the GATE and GATE-AFM EMA methods—trained and tested on PSBench—has been convincingly demonstrated through highly competitive results in the blind CASP16 experiment according to both external and our own assessments. Therefore, PSBench can fill the gap caused by the lack of large-scale EMA datasets and benchmarks in the field and can become a valuable resource for the community to develop more advanced machine learning-based EMA methods.
>
> **Reference**
>
> [1] Alisia Fadini, Recep Adiyaman, Shaima N. Alhaddad, Behnosh Behzadi, Jianlin Cheng, Xinyue Cui, Nicholas S. Edmunds, Lydia Freddolino, Ahmet G. Genc, Fang Liang, Dong Liu, Jian Liu, Quancheng Liu, Liam J. McGuffin, Pawan Neupane, Chunxiang Peng, David R. Shortle, Meng Sun, Haodong Wang, Qiqige Wuyun, Guijun Zhang, Xuanfeng Zhao, Wei Zheng, and Randy J. Read. "Highlights of Model Quality Assessment in CASP16." Proteins, accepted, 2025.

---

> > ### Comment · Area_Chair_maFG · 2025-08-06
> >
> > Reviewer DYzw - can you please check the rebuttal and respond?

---

> > ### Comment · Reviewer_DYzw · 2025-08-06
> >
> > The author's feedback basically answered my question, but  I will keep my score.

---

> > > ### Author Response · Authors · 2025-08-06
> > >
> > > Thank you for your response and for noting that our feedback addressed your original question. We would be grateful if you could share any remaining concerns that may have influenced your assessment. If there’s anything we can further clarify or improve upon, we would be happy to do so.

---

### Official Review · Reviewer_ZdDc · 2025-07-01

**Rating:** 4
**Confidence:** 4

**Summary:**

This paper addresses the Estimation of Model Accuracy task for protein complex structures by introducing PSBench, a large-scale, diverse, and standardized benchmark. PSBench comprises four sub-datasets drawn from the blind-prediction stages of CASP15 and CASP16, totaling over one million structures generated by deep-learning methods such as AlphaFold2-Multimer and AlphaFold3. Each model is annotated with ten complementary quality scores, covering global, interface and local metrics. The suite includes six baseline EMA methods, multiple evaluation metrics (Pearson/Spearman correlation, ranking loss, AUROC), and automated evaluation scripts. The authors train a graph-transformer EMA method (GATE) on PSBench and demonstrate its state-of-the-art performance in the CASP16 EMA competition, confirming PSBench’s utility and effectiveness.

**Additional Feedback:**

1. Since the paper mentions future support for third-party model uploads, provide a preliminary API design or plugin example and clarify contributor incentives as well as quality-control standards.
2. In an appendix, report runtime and memory-usage comparisons of several EMA methods on different hardware configurations, so users in resource-constrained environments can make informed choices.
3. It would be nice to have a simple colab tutorial to briefly introduce this approach(testing some samples), making other researchers easily get their hands dirty on it.
4. It would be nice to introduce more about the trained model in the appendix.

**Dataset Code Accessibility:**

Yes

**Dataset Code Comments:**

The code and data are available, the GitHub repo is clearly written, providing details on reproducing the work.

**Ethical Considerations:**

No, there are no or only very minor ethics concerns

**Final Justification:**

Thanks for the rebuttal; my major concerns have been addressed and largely resolved. However, as there seem to be some important details yet to be added to the appendix later( and still need to be justified and reviewed), I would retain my score.

**Limitations Weaknesses:**

1. Real PDB entries often include missing residues, ligands, or metal ions. It is unclear whether PSBench filters out, completes, or ignores these cases. As it would somehow affect other people reusing this work. A section on “preprocessing and quality control for missing atoms/residues” is recommended.
2. Ten metrics are provided, but there is no statistical analysis of their intercorrelations. Without this, it is unclear which metrics add unique information. A “metric redundancy analysis” and consideration of dimensionality reduction or weighted metrics during training would be beneficial.
3. Metrics like ICS, IPS, and QS are sensitive to protein–protein contact density, which varies greatly between targets. The paper does not discuss how interface density affects these scores. Adding this analysis would assess method reliability on weak-interaction interfaces.
4. The threshold used to binarize “high-quality vs. low-quality” models for AUROC calculation (e.g. TM-score ≥ 0.5) is not specified, nor is any analysis of how different thresholds affect ROC curves. The authors should detail threshold choices and include ROC comparisons under multiple cutoffs in the appendix.
5. Results are reported as average metrics, but no case studies examine targets with large prediction errors (e.g., severe TM-score under/overestimation). A “failure modes” subsection that dissects specific examples, such as heavily conformational targets or low-resolution templates, would illuminate method limitations and suggest directions for improvement.

**Strengths Contributions:**

1. PSBench collects four sub-datasets, covering 79 complex targets, 25 stoichiometries, 21 functional classes, sequence lengths from 96 to 8,460 residues, and over 1,000,000 models in total. It is a comprehensive collection in the field.
2. Each model is labeled with ten scores, including four TM-score variants and RMSD for global accuracy; lDDT for local accuracy; and interface metrics such as ICS, IPS, QS, and DockQ_wave.
3. Models are generated under authentic blind-prediction conditions (CASP15/16), distinguishing PSBench from benchmarks built on already-solved structures and better reflecting real-world EMA applications.
4. The benchmark incorporates six representative EMA approaches and four classes of evaluation metrics, enabling direct comparison of new and existing methods, quantification of performance gains, and reproducible evaluation pipelines via provided scripts.
5. A graph transformer(GATE) trained on PSBench achieves top-tier results in the CASP16 EMA challenge, and ranks highly on Pearson/Spearman correlations, ranking loss, and AUROC, showing PSBench’s training and testing value.
6. The code structure is clear, and the ReadMe provides a detailed instruction on how to reproduce the results.
7. The manuscript is clearly written, and the tables and figures are captioned properly.

---

> ### Author Rebuttal · Authors · 2025-07-30
>
> Thank you very much for the comprehensive and constructive assessment of our manuscript. Below is the point-by-point response to each comment/feedback.
>
> ---
>
> > **Comment 1: Real PDB entries often include missing residues, ligands, or metal ions. It is unclear whether PSBench filters out, completes, or ignores these cases. As it would somehow affect other people reusing this work. A section on “preprocessing and quality control for missing atoms/residues” is recommended.**
>
> **Response:**
>
> Thank you for the great suggestion. According to your recommendation, we prepared the following section on “Preprocessing and quality control for missing atoms/residues” to be added into the Appendix of the final manuscript.
>
> The positions of residues in all native PDB structures were reindexed to match with the full-length protein sequences of the corresponding CASP targets. Non-protein components, such as ligands and metal ions, were excluded. We then applied our annotation pipeline, which uses OpenStructure to align predicted and native structures based on sequence identity. When residues are missing from the native PDB structure, the corresponding residues in the predicted model are automatically excluded from evaluation by the OpenStructure alignment algorithm. This approach ensures that scoring is limited to the regions resolved in the native structures.
>
> ---
>
> > **Comment 2: Ten metrics are provided, but there is no statistical analysis of their intercorrelations. Without this, it is unclear which metrics add unique information. A “metric redundancy analysis” and consideration of dimensionality reduction or weighted metrics during training would be beneficial.**
>
> **Response:**
>
> We fully agree that analyzing inter-metric correlations is essential to distinguish unique from redundant metrics.
>
> To address this, we computed Pearson correlations among ten metrics across all PSBench datasets. Several metrics showed high redundancy—for example, ICS, ICS_Precision, and ICS_Recall (r ≥ 0.9), as well as QS_Best and QS_Global (r = 0.94)—due to overlapping focus on interface contacts. In contrast, LDDT, which evaluates atomic-level accuracy, had weaker correlations with interface and global fold metrics (e.g., r = 0.59 with DockQ_Wave, r = 0.50 with RMSD), suggesting it captures distinct information. IPS, based on the Jaccard index, showed moderate correlations (e.g., r ≈ 0.8 with ICS_Recall, r = 0.59 with RMSD), indicating partial redundancy with added value.
>
> Our new Principal Component Analysis (PCA) further clarified metric relationships. PC1 was driven by ICS, QS_Best, and QS_Global (loadings ≈ -0.35), representing interface quality. LDDT dominated PC2 (loading = -0.83), capturing atomic-level accuracy. RMSD and tmscore_mmalign defined PC3 and PC4, and IPS emerged as the primary contributor to PC5 (loading = 0.79).
>
> These findings confirm that while some metrics are somewhat redundant, others provide complementary insights. We will include the correlation and PCA results in the Appendix (“Metric redundancy analysis”) and propose a weighting scheme that down-weights highly correlated metrics.
>
> ---
>
> > **Comment 3: Metrics like ICS, IPS, and QS are sensitive to protein–protein contact density, which varies greatly between targets. The paper does not discuss how interface density affects these scores. Adding this analysis would assess method reliability on weak-interaction interfaces.**
>
> **Response:**
>
> Thank you for the great suggestion regarding the potential relationship between protein-protein contact density and several interface quality scores. To investigate this, we calculated the variance of ICS, IPS, and QS_Best scores for the CASP community structural models of each target. We then analyzed how these variances of the scores correlate with the protein-protein contact interface density of the native PDB structures.
>
> The Pearson correlation coefficients between interface density and score variance were consistently negative across all three metrics: −0.125 for ICS, −0.126 for IPS, and −0.115 for QS_Best. Although the correlations are modest in magnitude, they do suggest a weak inverse relationship—i.e., targets with lower contact density tend to exhibit higher variance in model quality scores.
>
> These findings align with your intuition that targets with weak interaction interface may present greater difficulty for structure prediction methods, resulting in less consistent model quality across methods. We will include this analysis in the final version of the manuscript.
>
> ---
>
> > **Comment 4: The threshold used to binarize “high-quality vs. low-quality” models for AUROC calculation (e.g. TM-score ≥ 0.5) is not specified, nor is any analysis of how different thresholds affect ROC curves. The authors should detail threshold choices and include ROC comparisons under multiple cutoffs in the appendix.**
>
> **Response:**
>
> Thank you for the great suggestion.
>
> As noted, we originally used the 75th percentile of the ground-truth scores (i.e., CASP official scores) for each target to define "high-quality" models in a relative sense. This percentile-based threshold adapts to the difficulty of individual targets and ensures balanced positive/negative class distribution for AUROC calculations across targets.
>
> According to your suggestion, we have now included additional AUROC results using fixed thresholds commonly used in the literature to determine “high-quality” and “low-quality” models. For example: we used TM-score ≥ 0.5 and DockQ_wave ≥ 0.49 to define high-quality structures based on the fixed thresholds. Under these fixed criteria, GATE-AFM achieved AUROC scores of 0.635 (TM-score) and 0.684 (DockQ_Wave), which is ranked second among all the methods. We will include a detailed comparative analysis using multiple thresholds in the Appendix of the final manuscript.
>
> ---
>
> > **Comment 5: Results are reported as average metrics, but no case studies examine targets with large prediction errors (e.g., severe TM-score under/overestimation). A “failure modes” subsection that dissects specific examples, such as heavily conformational targets or low-resolution templates, would illuminate method limitations and suggest directions for improvement.**
>
> **Response:**
>
> We fully agree that investigating targets with large prediction discrepancies can offer valuable insight into method limitations.
>
> Following your suggestion, we conducted a failure mode analysis. For targets like H1202 and T1206o, EMA methods show high MAE (mean average error) despite the predicted models being generally good and similar. In these cases, GATE-AFM preserves relative rankings (low ranking loss) but underestimates absolute TM-scores, indicating a calibration issue. Its focus on pairwise relationships can compress score predictions, reducing absolute accuracy.
>
> In contrast, H1265 includes only low-quality, diverse models with no resemblance to the native structure. Here, structural similarity among poor models misleads GATE-AFM, causing TM-score overestimation due to an uninformative similarity graph.
>
> These cases reveal two distinct failure modes: score compression in high-quality model sets and misleading graph structures in poor-quality sets. By comparison, targets like H1215 and H1223, with varied and well-formed models, show low MAE and strong performance.
>
> We will include these findings in a new “Failure Mode Analysis” subsection in the appendix.
>
> ---
>
> > **Additional Feedback 1 on third-party model uploads**
>
> **Response:**
>
> To support third-party contributions to PSBench, we will provide a RESTful API for submitting structural models in standard formats (e.g., PDB) along with required metadata.
>
> Users can submit models via a POST request with fields such as target_id, source_method (e.g., AlphaFold2/3), submitter_info, and structure files—both the predicted model and corresponding native structure. Bulk submission in compressed format will be supported. Native structures can be uploaded directly or specified via a recognized identifier (e.g., PDB ID), which the system will use to retrieve the reference structure automatically.
>
> All submissions will be stored with metadata to generate standardized scores.  The contributors and their related publications will be acknowledged in future PSBench releases.
>
> Submissions will be validated for structure format and metadata. Annotation will proceed only if a matching native structure is available and sequence/stoichiometry are compatible. Details will be provided in the “Preprocessing and Quality Control” section of the final manuscript.
>
> ---
>
> > **Additional Feedback 2 on runtime and memory-usage**
>
> **Response:**
>
> We appreciate your great suggestion.
>
> According to your suggestion, we have compared the runtime and memory-usage of several EMA methods on the computer system detailed in Appendix C.2. Some more accurate baseline EMA methods such as GATE, GATE-AFM and PSS are more memory-intensive and runtime-intensive, while some less accurate EMA methods such as GCPNet-EMA and DProQA require less memory and time. We will add these results into the Appendix of the final manuscript.
>
> ---
>
> > **Additional Feedback 3 on Colab tutorial**
>
> **Response:**
>
> In response to your feedback, we developed a Colab notebook that allows researchers to replicate our experiments and evaluate their own EMA methods. Users can upload predicted quality score CSVs for any number of targets and benchmark their results against PSBench EMA methods. This browser-based tool requires no local setup and makes PSBench more accessible. The notebook will be included in the PSBench GitHub repository in the final version.
>
> ---
>
> > **Additional Feedback 4 on more details about the trained model**
>
> **Response:**
>
> According to your suggestion, we will include a detailed description of the trained model of the GATE and GATE-AFM EMA methods in the Appendix of the final manuscript.
>
> ---

---

> > ### Comment · Reviewer_ZdDc · 2025-08-03
> >
> > Thanks for the rebuttal; my major concerns have been addressed and largely resolved. However, as there seem to be some important details yet to be added to the appendix later( and still need to be justified and reviewed), I would retain my score.

---

> > > ### Author Response · Authors · 2025-08-04
> > >
> > > We're pleased to know that your major concerns have been addressed and largely resolved. Thank you for letting us know more details about some comments are needed. Here we would like to provide more details about Comment 4 and Additional Feedbacks 2 and 4 that we feel did not have sufficient details before for you to assess.
> > >
> > > > **Comment 4 on the threshold used to binarize “high-quality vs. low-quality” models for AUROC calculation**:
> > >
> > > We added the AUROC scores of the EMA methods on the CASP16_inhouse_TOP5_dataset, under the commonly used TM-score threshold of 0.5 and the commonly used DockQ_wave threshold of 0.49, into Table 1 in the main manuscript, as shown in the table below. In terms of TM-score, The AUROC of GATE-AFM ranks second overall and significantly outperforms all other EMA methods (marked with *) except for PSS at threshold 0.5. In terms of DockQ_wave, at threshold 0.49, the AUROC of GATE-AFM also ranks second.
> > >
> > > |Methods|TM-score|||||DockQ_wave|||||
> > > |-|-|-|-|-|-|-|-|-|-|-|
> > > ||Corr_P|Corr_S|Loss|AUROC (75th)|AUROC (0.5)|Corr_P|Corr_S|Loss|AUROC (75th)|AUROC (0.49)|
> > > |GATE-AFM|_0.372_|**0.283**|**0.102**|**0.658**|_0.635_|**0.431**|**0.322**| **0.138**|**0.662**|_0.684_|
> > > |AFM-Confidence|0.259*|0.143*|_0.106_|0.597*|0.585*|0.252*|0.114*|0.151|0.593*|0.673|
> > > |PSS|**0.394**|_0.261_|0.114|_0.647_|**0.650**|_0.369_|_0.284_|_0.154_|_0.645_|**0.685**|
> > > |GCPNet-EMA|0.360|0.249|0.135|0.643|0.606*|0.355|0.264|0.169|0.648|0.672|
> > > |VoroMQA-dark|0.039*|0.144|0.129|0.609|0.570*|-0.013*|0.146*|0.163|0.622|0.588*|
> > > |VoroIF-GNN-pCAD-score|0.073*|0.105*|0.167*|0.589*|0.541*|0.074*|0.137*|0.204|0.615|0.570*|
> > > |VoroIF-GNN-score|0.065*|0.116*|0.193*|0.599*|0.565*|0.114*|0.170*|0.207*|0.622|0.601*|
> > > |DProQA|-0.051*|0.011*|0.194*|0.569*|0.543*|0.032*|0.071*|0.223*|0.587*|0.618|
> > >
> > > > **Additional Feedback 2 on runtime and memory-usage:**
> > >
> > > We have compared the runtime and memory-usage of the EMA methods in PSBench on the computer system detailed in Appendix C.2 (see the result in the table below). For GATE-AFM and GATE, the inference time after generating quality features for each model using external EMA methods is presented. The information for AFM-Confidence is not included because it is available with the models generated by AlphaFold. According to the results, multi-model EMA methods such as GATE, GATE-AFM, and PSS generally offer higher accuracy but are more computationally demanding. In contrast, single-model methods such as GCPNet-EMA and DProQA are more lightweight but typically achieve lower predictive accuracy.
> > >
> > > |EMA Method|Peak CPU Memory Usage|GPU Memory Usage|Run Time (min)|
> > > |-|-|-|-|
> > > |GATE-AFM (Inference only)|100947.91 MB|11868 MB|12.98|
> > > |GATE (Inference only)|100058.29 MB|11990 MB|31.04|
> > > |PSS|525.89 MB|N/A|18.88|
> > > |GCPNet-EMA|526.16 MB|643 MB|11.13|
> > > |VoroMQAs (CPU)|526.17 MB|N/A|19.28|
> > > |DProQA|526.16 MB|2629 MB|4.72|
> > >
> > > > **Additional Feedback 4 on more details about the trained model of GATE and GATE-AFM:**
> > >
> > > To predict the quality scores for a set of complex structural models of a protein, GATE and GATE-AFM first construct a pairwise model similarity graph in which each node represents a model, and an edge is formed between two nodes if the TM-score between the two corresponding models exceeds 0.5. Each node is annotated with features that include the average pairwise similarity scores between the model denoted by the node and other models, such as TM-score, QS-score, DockQ, and CAD-score, as well as single-model quality scores, including ICPS, EnQA, DProQA, and VoroMQA. The edge features capture the relationship between two connected models measured by multiple pairwise similarity scores, including TM-score, QS-score, DockQ, and CAD-score.
> > >
> > > From this full graph, a large number of subgraphs (typically 2000 to 3000) are sampled. Within each subgraph, node and edge features are embedded using an embedding module and then updated through a series of graph transformer layers that include multi-head attention mechanisms and feed-forward neural networks. A multilayer perceptron (MLP) then predicts a quality score for each node using the updated node embeddings. The only difference between GATE and GATE-AFM is in the initial node features, where GATE-AFM incorporates additional AlphaFold2-Multimer model score features as node features.
> > >
> > > The graph transformer is trained using a weighted loss function that combines the mean squared error (MSE) between the predicted score and the true quality score of a model, along with the mean absolute error (MAE) between the predicted score difference of each pair of models and the true score differences of the pair (denoted as pairwise loss). The search space of the hyperparameters is listed in Table S11 in Appendix B. The weights of the trained models are released at the GitHub repository.
> > >
> > > We hope the additional information above fully addresses your comments. Please let us know if you need more details on any point. Thank you very much.

---

> > > > ### Comment · Area_Chair_maFG · 2025-08-06
> > > >
> > > > Reviewer ZdDc - can you please check the additional comments and respond?

---

### Official Review · Reviewer_b6h4 · 2025-07-03

**Rating:** 5
**Confidence:** 2

**Summary:**

The paper introduces an extensive database of quality scores for protein structure predictions. This dataset can be used for training estimation of model accuracy (EMA) methods which can assess the quality of protein structure predictions without ground truths.

The novelty lies in the collection and calculation of a large number of quality scores on CASP15 and CASP16 data using the submitted predictions and the ground truths determined afterwards using experimental methods.

This new dataset is also used to train an EMA model using CASP15 data and evaluated on CASP16 data and it is shown that accurate estimation of model accuracy can be achieved using this dataset.

This reviewer has not worked in this field since 15 years, so please regard other reviewers' opinions more for evaluating this paper.

**Dataset Code Accessibility:**

Yes

**Dataset Code Comments:**

I was able to access the links.

**Ethical Considerations:**

No, there are no or only very minor ethics concerns

**Final Justification:**

I read the response of the authors and found it satisfactory. I raised my score. Maybe one final comment is it would be beneficial to talk about experimental methods for native structure (ground truth) prediction (such as x-ray crystallography) and whether there is any possibility of errors in them and how those could be mitigated.

**Limitations Weaknesses:**

Possible limitations:
1. The dataset uses only a limited number of protein targets (79) and using a larger one would help improve the results. The authors propose to extend it further, so it is a plus.
2. The novelty of the paper can be seen as limited, since it does not introduce a new quality metric or does not collect new predictions, but it uses existing predictions and targets from CASP15 and CASP16.

**Strengths Contributions:**

The paper has these strengths:
1. It addresses a critical gap in research to enable EMA prediction training.
2. Uses a large amount of training data.
3. The predictions in the training data are obtained by blind prediction methods since the structures were not known at the time of submission.
4. The protein targets were chosen using some important criteria to cover various protein types.
5. A comprenhesive set of quality metrics were used.

---

> ### Author Rebuttal · Authors · 2025-07-30
>
> Thank you very much for the comprehensive and constructive assessment of our manuscript. Below is the point-by-point response to each comment.
>
> ---
>
>
>
> > **Comment 1: The dataset uses only a limited number of protein targets (79) and using a larger one would help improve the results. The authors propose to extend it further, so it is a plus.**
>
> **Response:**
>
>
>
> Thank you for the insightful comments on the number of protein complex targets currently available in PSBench. Even though we have shown that only two CASP15 datasets associated with 31 / 40 CASP15 targets have enabled us to develop and train state-of-the-art EMA methods (GATE and GATE-AFM), it would be very useful to add more structural models of more protein complex targets into PSBench to better train large, advanced EMA models.
>
>
>
> Therefore, we have collected about 4,500 protein targets whose native structures were released between the end of CASP16 (August 2024) and July 2025 to generate new structural models using AlphaFold3. Because the native structures of these targets were released after the cut-off date of the training data of AlphaFold3, the model generation mimics the blind prediction setting. We have started to generate about 200 models per target. The generated models will be compared with the native structures to assign quality scores using the model annotation pipeline in PSBench. The model generation and annotation process is expected to be completed in one and a half months. The new structural models and their quality scores will be added into the final data repository of PSBench. This addition will greatly expand the structural space covered by PSBench and provide about 900,000 new structural models of many different proteins to train and test EMA methods. The description of the new dataset will be added into the final version of this manuscript.
>
> Moreover, to support third-party contributions to PSBench, we will provide a RESTful API for submitting structural models in standard formats (e.g., PDB) along with required metadata. Users can submit models via a POST request with fields such as target_id, source_method (e.g., AlphaFold2/3), submitter_info, and structure files—both the predicted model and corresponding native structure. Bulk submission in compressed format will be supported. Native structures can be uploaded directly or specified via a recognized identifier (e.g., PDB ID), which the system will use to retrieve the reference structure automatically. All submissions will be stored with metadata to generate standardized scores. The contributors and their related publications will be acknowledged in future PSBench releases. We will add this as a subsection into the Appendix of the final version of this manuscript.
>
> Furthermore, we will continue to incorporate structural models from future blind CASP experiments such as the upcoming CASP17 to be held in 2026 into PSBench.
>
> We are confident the efforts above will address the concern about the limited number of targets effectively.
>
> ---
>
> > **Comment 2: The novelty of the paper can be seen as limited, since it does not introduce a new quality metric or does not collect new predictions, but it uses existing predictions and targets from CASP15 and CASP16.**
>
> **Response:**
>
>
>
> Thank you for the valuable comment. As mentioned in our response to the previous comment, we have collected about 4,500 protein targets whose native structures were released between the end of CASP16 (August 2024) and July 2025 to generate new structural models using AlphaFold3. Because the native structures of these targets were released after the cut-off date of the training data of AlphaFold3, the model generation mimics the blind prediction setting. We have started to generate about 200 models per target. The generated models will be compared with the native structures to assign quality scores using the model annotation pipeline in PSBench. The model generation and annotation process is expected to be completed in one and a half months. The new structural models and their quality scores will be added into the final data repository of PSBench. This addition will greatly expand the structural space covered by PSBench and provide about 900,000 structural models of many different proteins to train and test EMA methods. It will enhance the novelty of this work. The description of the new dataset will be added into the final version of this manuscript.
>
>
> As for evaluation metrics, we used 10 well-established, complementary metrics that are widely used in the field and by CASP assessor, which provide a comprehensive assessment of the quality of structural models. Therefore, we rely on them without introducing a new metric that may not be adopted by the community. However, if some new metric emerges in the field, we will add it into PSBench as soon as possible.
>
>
>
> ---

---

> > ### Comment · Area_Chair_maFG · 2025-08-06
> >
> > Reviewer b6h4 - can you please check the rebuttal and respond?

---

### Official Review · Reviewer_Jg2y · 2025-07-05

**Rating:** 5
**Confidence:** 3

**Summary:**

This paper introduces PSBench, a comprehensive benchmark suite for training and evaluating methods that estimate the accuracy of protein complex structural models (EMA methods). PSBench comprises four large-scale datasets with over one million structural models from CASP15 and CASP16 competitions, covering 79 diverse protein complex targets. Each model is annotated with 10 complementary quality scores at global, local, and interface levels. The authors demonstrate PSBench's utility by training GATE, a graph transformer-based EMA method, which achieved top performance in the blind CASP16 competition.

**Additional Feedback:**

The paper would benefit from:
1. More detailed analysis of potential biases in the AlphaFold-dominated model generation
2. Discussion of computational requirements for training EMA methods on the full datasets
3. Clearer guidelines for practitioners on selecting appropriate quality scores for different applications

**Dataset Code Accessibility:**

Yes

**Dataset Code Comments:**

The paper provides excellent accessibility with datasets available on Harvard Dataverse and source code on GitHub. The documentation includes comprehensive reproduction instructions and system requirements.

**Ethical Considerations:**

No, there are no or only very minor ethics concerns

**Final Justification:**

Authors rebuttal clarified my major concerns.

**Limitations Weaknesses:**

1. Despite including 79 targets, this may not fully capture the complete diversity of protein complexes. The authors acknowledge this limitation and plan future expansions.
2. The datasets are limited to CASP15 and CASP16 timeframes. While this provides a natural train/test split, broader temporal coverage would strengthen the benchmark.
3. The structural models are generated primarily by AlphaFold2-Multimer and AlphaFold3, which may limit the diversity of prediction approaches represented in the datasets.
4. Some implementation details and edge cases in the annotation pipeline could be better documented to ensure reproducibility across different research groups.

**Strengths Contributions:**

1. PSBench represents a significant advancement in dataset scale for protein complex EMA research, providing over one million labeled structural models compared to previous datasets with tens of thousands.
2. The datasets are generated from truly blind prediction settings during CASP competitions, where true structures were unavailable during prediction. This provides more realistic training and evaluation conditions compared to simulated prediction environments.
3. The inclusion of 10 complementary quality scores spanning global, local, and interface accuracy measures provides a multifaceted view of model quality, enabling more nuanced EMA method development.
4. PSBench includes baseline EMA methods, standardized evaluation metrics, and automated evaluation tools, making it immediately usable for the research community.
5. The successful application of GATE trained on PSBench data, achieving top performance in CASP16, validates the benchmark's practical utility for advancing state-of-the-art methods.

---

> ### Author Rebuttal · Authors · 2025-07-30
>
> Thank you very much for the valuable comments on the strengths and limitations of our manuscript. Below is the point-by-point response to each comment.
>
> ---
>
> > **Comment 1: Despite including 79 targets, this may not fully capture the complete diversity of protein complexes. The authors acknowledge this limitation and plan future expansions.**
>
> **Response:**
>
> Thank you for the insightful comments on the number and diversity of protein complex targets currently available in PSBench. Even though we have shown that only two CASP15 datasets associated with 31 / 40 CASP15 targets have enabled us to develop and train state-of-the-art EMA methods (GATE and GATE-AFM), we fully agree with you that it is very useful to add more structural models of more protein complex targets into PSBench to increase its diversity and better train/evaluate EMA methods.
>
> Therefore, we have collected about 4,500 protein targets whose native structures were released between the end of CASP16 (August 2024) and July 2025 to generate new structural models using AlphaFold3. Because the native structures of these targets were released after the cut-off date of the training data of AlphaFold3, the model generation mimics the blind prediction setting. We have started to generate about 200 models per target. The generated models will be compared with the native structures to assign quality scores using the model annotation pipeline in PSBench. The model generation and annotation process are expected to be completed in one and a half months. The new structural models and their quality scores will be added into the final data repository of PSBench. This addition will greatly increase the diversity of protein complexes covered by PSBench and provide about 900,000 new structural models of many diverse proteins to train and test EMA methods. The description of the new dataset will be added into the final version of this manuscript.
>
> Moreover, to support third-party contributions to PSBench, we will provide an API for users to submit structural models to PSBench, which will be annotated (labeled) by the model annotation pipeline in PSBench. The contributors and their related publications will be acknowledged in future PSBench releases. Furthermore, we will continue to incorporate the models of future CASP experiments such as CASP17 to be held in 2026 into PSBench.
>
> We are confident the efforts above will address the concern about the diversity of targets effectively.
>
> ---
>
> > **Comment 2: The datasets are limited to CASP15 and CASP16 timeframes. While this provides a natural train/test split, broader temporal coverage would strengthen the benchmark.**
>
> **Response:**
>
> We fully agree with you that it is important to add more protein complex targets into PSBench to provide a broader temporal coverage of protein complex structures. As described in our response to Comment 1 above, we have addressed this issue using multiple measures to strengthen PSBench. Particularly, we have started to generate and annotate structural models for about 4,500 protein targets whose true/native structures are released after the end of CASP16, which will be done in one and a half months. This addition will greatly broaden the temporal coverage of the benchmark.
>
> ---
>
> > **Comment 3: The structural models are generated primarily by AlphaFold2-Multimer and AlphaFold3, which may limit the diversity of prediction approaches represented in the datasets.**
>
> **Response:**
>
> We apologize that we did not make it clear that the two CASP community datasets (CASP15_community_dataset and CASP16_community_dataset) in PSBench include both the models generated by AlphaFold2-Multimer/AlphaFold3 and some other non-AlphaFold structure predictors. The structural models in CASP15_community_dataset and CASP16_community_dataset were generated by 84/88 diverse predictors from different research groups in the world using a variety of structure prediction tools and strategies including both AlphaFold2-Multimer/AlphaFold3 and other predictors such as ESMFold, RosettaFold, and complex docking methods. Therefore, these two datasets capture the diversity of prediction approaches in the field. We will add this description in the final version of the main manuscript.
>
> We also recognize that new methods may continue to emerge in the field. We plan to incorporate models generated by them in future PSBench releases to further expand the diversity of prediction approaches represented in the benchmark.  We will add this point to the final version of the manuscript.
>
> ---
>
> > **Comment 4: Some implementation details and edge cases in the annotation pipeline could be better documented to ensure reproducibility across different research groups.**
>
> **Response:**
>
> Thank you for highlighting the need for clearer documentation of implementation details and edge cases in our annotation pipeline. While the pipeline, based on OpenStructure, is robust, certain edge cases during large-scale processing warrant explicit description. The implementation details/edge cases below will be added to the Appendix and GitHub repository of the final manuscript.
>
> Some structural models in the CASP community datasets failed annotation due to format issues or violations of OpenStructure’s assumptions. For example, non-monotonic residue numbering (e.g., H1272TS191_1 from CASP16) caused alignment failures, and malformed PDB files with duplicate atom labels (e.g., H1114TS229_1 from CASP15) triggered parsing errors.
>
> If OpenStructure failed but US-align succeeded, we retained the model and reported TM-score from US-align. If both failed, the model was excluded. These failures were rare, affecting fewer than 20 out of over one million models.
>
> Documenting these details above will improve transparency and ensure reproducibility of the annotation pipeline.
>
> ---
>
> > **Additional feedback 1 on more detailed analysis of potential biases in the AlphaFold-dominated model generation.**
>
> **Response:**
>
> Thank you for raising the important issue of potential bias from AlphaFold-dominated model generation in PSBench. While PSBench includes CASP15 and CASP16 community datasets featuring models from both AlphaFold and various non-AlphaFold predictors (see response to Comment 3), most structural models in PSBench were still generated by AlphaFold. This reflects the current dominance of AlphaFold in the field but does introduce method-specific bias.
>
> To assess this bias, we compared TM-score distributions between community datasets (CASP15_community_dataset and CASP16_community_dataset) consisting of models generated by both AlphaFold and other non-AlphaFold methods and in-house datasets (CASP15_inhouse_dataset and CASP16_inhouse_dataset) consisting of models generated by AlphaFold only. The community datasets, with 23,841 models from diverse methods, showed a lower average TM-score (0.7166 vs. 0.7873) and higher standard deviation (0.2394 vs. 0.2160) than the AlphaFold-only in-house datasets, indicating slightly lower average quality and greater variation.
>
> To examine the impact of the bias on EMA performance, we evaluated GATE (trained on the CASP15 community data) and GATE-AFM (trained on the CASP15 in-house data) using the CASP16_inhouse_TOP5_dataset, a high-confidence AlphaFold in-house model subset. GATE-AFM outperformed GATE in Spearman's correlation, loss, and AUROC for both TM-score and DockQ_Wave, while GATE achieved slightly higher Pearson's correlation on TM-score.
>
> These results show that model generation bias in training data can affect generalization. We will include this analysis in the Appendix of the final manuscript.
>
> ---
>
> > **Additional feedback 2 on the discussion of computational requirements for training EMA methods on the full datasets.**
>
> **Response:**
>
> The computational cost of training EMA methods on the full PSBench dataset depends heavily on how structural models are used as inputs. For single-model EMA methods, where each model is processed independently, the training time scales linearly with the number of models (e.g., 10,942 models in the CASP15_community_dataset). In contrast, multi-model methods like GATE construct a pairwise model similarity graph for each target using all models of the target available in the dataset, which introduces quadratic or higher time complexity with respect to the number of models per target. However, after the similarity graph is constructed, extracting subgraphs and using them for training is faster. Under this setup, GATE was trained using a single A100 GPU (80 GB) and required approximately 2 minutes per epoch. We will add the discussion of the computational requirements into the Appendix of the final manuscript.
>
> ----
>
> > **Additional feedback 3 on the clearer guidelines for practitioners on selecting appropriate quality scores for different applications.**
>
> **Response:**
>
> We will add the following detailed guidance linking each quality score to relevant structural biology and bioinformatics applications in the Appendix of the final manuscript.
>
> TM-score variants are ideal for evaluating global fold accuracy in tasks like structure-based function annotation and complex template retrieval, as they capture overall topology and are widely used in fold recognition benchmarks.
>
> RMSD is appropriate for atomic-level accuracy in applications such as structural refinement or drug binding site validation, where small positional deviations can impact function.
>
> For interface-focused tasks—such as docking, binder design, and epitope mapping—DockQ_Wave and IPS offer comprehensive assessments of contact and interface quality. ICS_Precision and ICS_Recall help evaluate specificity and coverage of predicted contacts. QS_Global measures overall interface accuracy, while QS_Best highlights the highest-quality interface, useful in asymmetric or functionally focused assemblies.
>
>
> ---

---

> > ### Comment · Reviewer_Jg2y · 2025-08-04
> >
> > Thank you for clarifications which addresses my major concerns. I changed my score to five.

---

> > > ### Author Response · Authors · 2025-08-04
> > >
> > > We are delighted to know our responses have addressed your concerns. Thank you so much for the highly valuable comments to help us improve the manuscript.

---

### Note · Authors · 2025-08-12

Dear Area Chair and Reviewers,

Thank you for reviewing our manuscript and providing valuable, constructive feedback throughout the process. This work **addresses a critical gap in protein structure prediction** — the absence of large-scale datasets and benchmarks enabling the AI community to train and rigorously evaluate machine learning methods for *estimating the accuracy of predicted protein complex structural models* (EMA), a capability much needed by the life science community.

We greatly appreciate that the reviewers recognized the **major strengths** of the work, which we believe make it a strong contribution to the NeurIPS 2025 Datasets and Benchmarks track:
* **Unprecedented scale and diversity** — over one million labeled models from 79 diverse protein complexes, spanning wide stoichiometries, functional classes, and sequence lengths.
* **Realistic blind-prediction data** — generated in CASP competitions with unknown true structures at prediction time, ensuring authentic evaluation conditions.
* **Rich multi-metric annotations** — ten complementary measures capturing global, local, and interface accuracy.
* **Comprehensive evaluation framework** — six baseline EMA methods, standardized metrics, automated tools, and reproducible scripts.
* **Proven impact** — models trained on PSBench achieved top-tier performance in CASP16, demonstrating its value for advancing EMA research.

We are grateful for the reviewers’ thoughtful comments and feedback on topics such as target coverage, temporal & methodological diversity, implementation details, potential biases, computational requirements, guidelines for method selection, dataset novelty, real-world data complexity, evaluation thresholds, failure modes, dataset expansion, tutorial support, trained model details, and the demonstration of EMA method effectiveness.

We have worked diligently during the rebuttal process to address all the points, including conducting new experiments and data analyses and refining the manuscript. We are pleased that **all the reviewers' concerns have been addressed**. We believe **the revisions have substantially strengthened the work**. All changes and new data will be incorporated into the final revised manuscript. If you need any additional information, please let us know.

We thank you for your guidance in improving this work, and we hope PSBench will serve as a valuable resource for the NeurIPS and broader scientific communities.

Sincerely,

Pawan, Jian, Jianlin

---

### Decision · Program_Chairs · 2025-09-18

**Decision:**

Accept (poster)

**Comment:**

(a) This paper introduces PSBench, a comprehensive benchmark suite for training and evaluating methods that estimate the accuracy of protein complex structural models (EMA methods). PSBench comprises 4 large-scale datasets generated during the CASP15 and CASP16 competitions, containing over 1M structural models covering 79 diverse protein complex targets. Each structural model is annotated with ten complementary quality scores at global, local, and interface levels, including TM-score variants, RMSD, lDDT, and interface-specific metrics like ICS, IPS, QS, and DockQ. The key finding is that PSBench enables the development of highly effective EMA methods. The authors demonstrate this by training GATE — a graph transformer-based EMA method — on CASP15 data. When blindly tested in CASP16, GATE ranked among the top-performing EMA methods out of 38 international predictors, validating PSBench's utility for advancing state-of-the-art EMA research.

(b) The reviewers consistently recognised several significant strengths of PSBench. PSBench represents a major advancement in dataset scale for protein complex EMA research, providing over one million labeled structural models compared to previous datasets with tens of thousands. The benchmark covers 79 diverse protein complex targets with 25 stoichiometries and 21 functional classes (reviewer Jg2y, reviewer ZdDc). The datasets are generated from truly blind prediction settings during CASP competitions, where true structures were unavailable during prediction. This provides more realistic training and evaluation conditions compared to simulated prediction environments (reviewer Jg2y, reviewer ZdDc). The inclusion of ten complementary quality scores spanning global, local, and interface accuracy measures provides a multifaceted view of model quality, enabling more nuanced EMA method development (reviewer Jg2y, reviewer ZdDc). PSBench includes baseline EMA methods, standardised evaluation metrics, automated evaluation tools, and reproducible scripts, making it immediately usable for the research community (reviewer Jg2y, reviewer ZdDc). The successful application of GATE trained on PSBench data, achieving top performance in CASP16, validates the benchmark's practical utility for advancing state-of-the-art methods (all reviewers). The paper provides excellent accessibility with datasets available on Harvard Dataverse and source code on GitHub, with comprehensive reproduction instructions and clear documentation (reviewer Jg2y, reviewer ZdDc).

(c) Although it covers a large number targets, it still doesn’t capture the full diversity of protein complexes (but perhaps can’t hope to?) . The temporal coverage is also limited to CASP15 and CASP16 timeframes (reviewer Jg2y, reviewer b6h4). The structural models are generated primarily by AlphaFold2-Multimer and AlphaFold3, which may limit the diversity of prediction approaches represented in the datasets (reviewer Jg2y). Some implementation details and edge cases in the annotation pipeline could be better documented. Issues like handling of missing residues, ligands, or metal ions in PDB entries were not clearly addressed (reviewer ZdDc). The paper lacks statistical analysis of intercorrelations between the ten quality metrics, making it unclear which metrics add unique information (reviewer ZdDc). Results are reported as average metrics, but no case studies examine targets with large prediction errors or specific failure modes (reviewer ZdDc). The threshold used to binarise "high-quality vs. low-quality" models for AUROC calculation was not clearly specified (reviewer ZdDc).

(d) The recommendation to accept this paper is based on its substantial contribution to addressing a critical need in the protein structure prediction community and its demonstrated practical impact. The benchmark's unprecedented scale and authentic blind-prediction data from CASP competitions provide a robust foundation for EMA method development. The comprehensive annotation with ten complementary quality metrics enables multifaceted evaluation. The success of GATE in CASP16, where it ranked among the top EMA methods globally, provides compelling evidence of PSBench's effectiveness. This real-world validation demonstrates that the benchmark can indeed advance state-of-the-art methods.

(e) The discussion period was highly productive, with the authors providing detailed responses and concrete plans to address reviewer concerns. In response to concerns about limited target coverage (reviewer Jg2y, reviewer b6h4), the authors announced a major expansion effort. They have collected approximately 4,500 protein targets whose native structures were released between August 2024 and July 2025, and are generating about 200 models per target using AlphaFold3 (total ~900k). The authors clarified that the CASP community datasets include models from 84-88 diverse prediction methods beyond just AlphaFold variants, addressing concerns about methodological diversity (reviewer Jg2y). The authors provided extensive additional technical details about the GATE architecture, training procedures, hyperparameter search spaces, and evaluation protocols in response to reviewer ZdDc's requests for more implementation specifics. The authors outlined plans for a RESTful API that will allow third-party researchers to submit structural models for annotation, supporting community contributions and continuous benchmark expansion. The authors committed to adding detailed metric intercorrelation analysis, failure mode studies, and comprehensive documentation of preprocessing procedures to address reviewer ZdDc's concerns about technical completeness. The authors provided additional evidence of GATE's effectiveness, noting that their group was invited to present at the CASP16 meeting and contribute to the official CASP16 EMA assessment paper, with only five top EMA prediction groups worldwide receiving this recognition. Reviewer Jg2y noted that their "major concerns" were clarified and maintained their "Accept" rating. Reviewer b6h4 found the response satisfactory and raised their score to "Accept." Reviewer ZdDc acknowledged that their "major concerns have been addressed and largely resolved," though they maintained their "Borderline accept" rating pending the addition of promised technical details. Reviewer DYzw confirmed that the authors' feedback "basically answered" their question. The authors' comprehensive response, concrete expansion plans, and demonstrated real-world impact through CASP16 success strongly support the value and readiness of PSBench as a foundational resource for the protein structure prediction community.

===== FINAL UPDATE FROM DB Track PCs ====

The final decision for this paper has been taken by the program chairs after consultation with the SACs. All Senior Area Chairs have ranked papers according to the feedback from the AC during the review process. We decided to leave the original meta-review to reflect the opinion of the AC in light of the initial discussions with reviewers and SAC.